# Masked multi-prediction for multi-aspect anomaly detection

**Yassine Naji**                                            *yassine.naji@cea.fr*
*Université Paris-Saclay, CEA, List, 91120, Palaiseau, France*
*Université Paris-Saclay, CNRS, LISN, 91400, Orsay, France*

**Romaric Audigier**                                    *romaric.audigier@cea.fr*
*Université Paris-Saclay, CEA, List, 91120, Palaiseau, France*

**Aleksandr Setkov**                                      *aleksandr.setkov@cea.fr*
*Université Paris-Saclay, CEA, List, 91120, Palaiseau, France*

**Angélique Loesch**                                     *angelique.loesch@cea.fr*
*Université Paris-Saclay, CEA, List, 91120, Palaiseau, France*

**Michèle Gouiffès**                      *michele.gouiffes@universite-paris-saclay.fr*
*Université Paris-Saclay, CNRS, LISN, 91400, Orsay, France*

**Reviewed on OpenReview:** *https://openreview.net/forum?id=7wybYcK1pw*

## Abstract

In this paper, we address the anomaly detection problem in the context of heterogeneous normal observations and propose an approach that accounts for this heterogeneity. Although prediction-based methods are common to learn normality, the vast majority of previous work predicts a single outcome, which is generally not sufficient to account for the multiplicity of possible normal observations. To address this issue, we introduce a new masked multi-prediction (MMP) approach that produces multiple likely normal outcomes, and show both theoretically and experimentally that it improves normality learning and leads to a better anomaly detection performance. In addition, we observed that normality can be characterized from multiple aspects, depending on the types of anomalies to be detected. Therefore, we propose an adaptation (MMP-AMS) of our approach to cover multiple aspects of normality such as appearance, motion, semantics and location. Since we model each aspect separately, our approach has the advantage of being interpretable and modular, as we can select only a subset of normality aspects. The experiments conducted on several benchmarks show the effectiveness of the proposed approach.

## 1 Introduction

*Video anomaly detection (VAD)* is crucial for many applications such as video surveillance or autonomous driving for instance. However, it is still an open research problem due to several challenges. The first one is the *scarcity* of anomaly examples and the lack of their corresponding annotations. Indeed, by definition, anomalies are unexpected and usually diverse, therefore, it is infeasible to collect enough representative samples. This makes classical supervised methods ineffective due to the class imbalance issue. Thus, this problem is often considered from the One-Class perspective where a model of normality is learned from normal data only and detects anomalies as outliers.

Diverse approaches of modeling normal data exist in the One-Class anomaly detection literature. On the one hand, many existing methods model directly the distribution of normal data or a high-level representation of it, denoted by $\mathbb{P}(f(X))$ where $f$ is a transformation applied to normal data samples $X$. This category includes probabilistic methods such as diffusion models (Flaborea et al. (2023); Wyatt et al. (2022)) or GANs (Ravanbakhsh et al. (2017); Liu et al. (2018)) which learn a probabilistic model of normality. At inference,

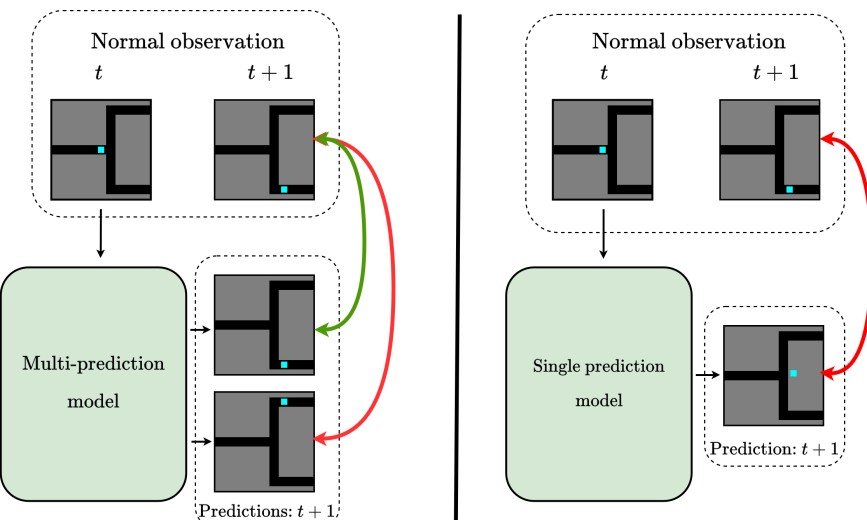

Figure 1: An example illustrating the motivation behind performing multi-prediction instead of a single prediction. We generated a synthetic dataset illustrating a road scenario where a car (in blue) is at an intersection, and can perform two actions (turning left or right) which generates two different next states. The right figure shows a single prediction network trained to perform next position prediction, while the left figure shows another one trained to perform multiple predictions using our proposed loss functions. Green arrow indicates a correct matching between the ground truth observation at time $t + 1$ and a model prediction, while red arrow indicates incorrect matching.

samples with low likelihood given the learned density function are considered as anomalies. Another popular family of approaches are distance-based methods (Ramachandra et al. (2022); Singh et al. (2023)). Those methods learn an embedding space and the corresponding metric, to ensure that abnormal data are far apart from the normal data. Other approaches perform clustering in some low dimensional space (Ionescu et al. (2019a;b); Wang et al. (2020)) to define normality regions, and the anomaly is deduced from the distance to them. Recent approaches such as Park et al. (2020); Gong et al. (2019); Liu et al. (2021); Bergaoui et al. (2022) model the normality via a discrete set of prototypes in the latent space. On the other hand, other approaches learn normal patterns by training a model on a pretext task, thereby learning features of normality such as appearance and motion (Georgescu et al. (2021a); Barbalau et al. (2022); Wang et al. (2022)). A pretext task usually consists of learning $\mathbb{P}(f(X)|g(X))$ where $g$ and $f$ are transformations of $X$, and $g(X)$ is informative about $f(X)$. A model is trained on normal data only via those pretext tasks. For a given test sample, the abnormality score can be inferred from the model's inability to perform the task correctly. There are two main pretext tasks used in the VAD literature: *reconstruction-based* (Gong et al. (2019); Bergaoui et al. (2022); Park et al. (2020); Georgescu et al. (2021b)) and *future prediction-based* (Liu et al. (2018); Naji et al. (2022); Liu et al. (2021); Dong et al. (2020); Nguyen & Meunier (2019); Tang et al. (2020)). Reconstruction-based approaches generally train a neural network which reproduces the normal training data from a low dimensional space. The fundamental assumption is that the model will not be able to generalize well to anomalies. Differently, future prediction-based approaches train a model to predict a future outcome given the past. At inference, the anomaly score is deduced from the prediction error.

The choice between reconstruction-based and prediction-based methods involves trade-offs. While the former reconstruct training data well, they also tend to reconstruct anomalies due to the generalization abilities of neural networks (Gong et al. (2019)). On the contrary, the latter has the advantage of predicting poorly anomalies, as the model cannot simply reproduce the input as with reconstruction-based methods. However, these methods have difficulty predicting normal future scenarios because of their diversity. Indeed, many existing future-prediction methods perform single prediction, which is often not enough to characterize all possible future outcomes (Babaeizadeh et al. (2018)). To illustrate this point, let us consider a road scenario (Figure 1), where the task consists in modeling the normal set of car trajectories. We assume that the car is

located at an intersection at time $t$, and can turn left or right at time $t + 1$. Note that both trajectories are normal in this case, but it is impossible to predict both possibilities with a single prediction. Even worse, this leads to an abnormal prediction (Figure 1 right), which is different from all normal trajectories. In this paper, we aim to solve this problem by better modeling the distribution of possible future scenarios. Thus, we propose a model that performs *multiple predictions* instead of a single one. In order to cover normal possibilities, we use the nearest neighbor loss (Guzmán-rivera et al. (2012); Bhattacharyya et al. (2018); Nguyen et al. (2018)), which is often used in the setting of multiple choice learning, and we introduce a new non-participation loss that ensures a balanced training of all predictors. Thus, our approach offers the advantages of prediction-based methods in terms of poor anomaly recovery, and enhances normality learning through multi-prediction, improving anomaly detection performance.

Another VAD challenge is that, in order to detect anomalies, it is necessary to determine normality. However, the definition of what is considered normal depends on the context and the application, which also influences the choice of normality aspects to be modeled (e.g., appearance, motion, etc.). While certain aspects of normality are not relevant to detect anomalies for some applications (e.g., the location of a person on a sidewalk when the objective is to detect violence), they tend to be crucial in others (e.g., the location of a person when the objective is to detect jaywalking). Therefore, we model several object-level aspects such as appearance, motion and class-semantics, as well as location-related anomalies. Our approach is interpretable and modular, since it assigns an anomaly score for each aspect. This allows us to adapt our method to applications that require only a subset of the aspects to be modeled while providing information about the anomaly type. In summary, our contributions are as follows: 1) a novel and generic *masked multi-prediction* (MMP) approach for anomaly detection in the context of heterogeneous normal data; 2) an adaptation of MMP to model multiple normality aspects (MMP-AMS); 3) a new *non-participation loss* to better model the multiplicity of normal scenarios; 4) a theoretical analysis and experiments showing the effectiveness of our methodology.

## 2 Related work

**Multi-prediction learning**: also known as multiple choice learning (Lee et al. (2016); Dey et al. (2017); Lee et al. (2017); Guzmán-rivera et al. (2012)) or multiple hypotheses learning (Rupprecht et al. (2017); Nguyen et al. (2018)), is a task where multiple models are learned to produce diverse predictions. During training, samples are assigned to the minimum loss predictor. This technique has been used for tasks involving aleatoric uncertainty such as future prediction (Bhattacharyya et al. (2018)), human pose estimation (Rupprecht et al. (2017)), image segmentation (Guzmán-rivera et al. (2012); Dey et al. (2017)). In the context of anomaly detection in images (a.k.a novelty detection), Nguyen et al. (2018) proposed a multiple hypothesis auto-encoder which performs multiple reconstructions. Differently from Nguyen et al. (2018), we perform *masked multi-prediction* by combining masking with multi-prediction in order to limit the capacity of the model to recover anomalies. We also propose a new *non-participation loss* that penalizes only the predictors that do not participate in training, and show both theoretically and empirically that it improves the coverage of normal possibilities. In addition, our framework learns multiple aspects of normality, enabling it to detect the corresponding anomaly types in the context of videos. To our knowledge, our work is the first to propose a multi-prediction approach for VAD.

**Masked prediction for anomaly detection**: the procedure of predicting a masked input has been studied in several applications of anomaly detection such as industrial inspection (Zavrtanik et al. (2021); Tong et al. (2023); Huang et al. (2022); Jiang et al. (2023)), where a model is trained to perform the inpainting task, hyperspectral imagery (Li et al. (2023)) or time series (Fu & Xue (2022)). In the context of VAD, most of previous studies perform a temporal masking by performing tasks such as future frame prediction (Liu et al. (2018; 2021); Dong et al. (2020); Tang et al. (2020)), middle frame prediction (Barbalau et al. (2022); Georgescu et al. (2021a)) or video event completion (Yu et al. (2020)). Barbalau et al. (2022) proposed a spatial masking procedure which consists of random patch inpainting. Differently from those studies, we propose a joint spatio-temporal masking procedure in order to learn correlations between the spatial and temporal patterns of normal data. Moreover, we propose to perform multiple predictions instead of a single prediction in order to better learn diverse normal patterns and therefore improve anomaly detection performance.

**Video anomaly detection**: most implicit VAD approaches use self-supervised learning to model normality. The model is trained on a given task using normal data only and it is expected not to generalize well to abnormal samples. Usually, these tasks are designed to characterize a certain aspect of normal data such as appearance or motion, which allows the model to detect the corresponding anomaly types. Hasan et al. (2016) were one of the first to propose a *reconstruction*-based method by training an auto-encoder to recover handcrafted appearance and motion features. As pointed out by Gong et al. (2019), auto-encoders are able to reconstruct anomalies due to the extrapolation capabilities of deep learning models, which is not suitable for distinguishing between normal and abnormal samples. The reconstruction task can be further constrained using a memory module as proposed in (Gong et al. (2019)). Georgescu et al. (2021b) proposed to train a model via an adversarial objective function, where normal data is well reconstructed and some pseudo-anomalies are explicitly misreconstructed. A major self-supervised paradigm to learn normality consists in training a model to perform *future frame prediction*. Liu et al. (2018) trained a generator using an adversarial objective function to predict a future frame and its optical flow given few past frames. Ravanbakhsh et al. (2017) trained two generators to perform image-to-image translation between RGB and optical flow modalities in order to learn both appearance and motion normality. Liu et al. (2021) introduced a hybrid framework for frame prediction and optical flow reconstruction at the object-level by making use of a pretrained object detector. During inference, the anomaly score is computed based on a sampled future object-level frame. However, one sample may not be representative of the full distribution of future scenarios. In order to cover various modes of this distribution, we propose to train our framework to produce diverse and representative predictions, using the nearest neighbor loss (Guzmán-rivera et al. (2012)) and our novel non-participation loss. Similarly to Liu et al. (2021), we propose to model normality at the object-level to provide better robustness to scene changes and background variety. Recent works introduced other self-supervised tasks for object-level normality learning such as spatio-temporal jigsaw puzzle (Wang et al. (2022)), video event completion (Yu et al. (2020)) or random patch inpainting (Barbalau et al. (2022)). Barbalau et al. (2022); Georgescu et al. (2021a); Shi et al. (2023) proposed to combine multiple tasks such as arrow-of-time prediction, motion irregularity, middle-frame prediction and knowledge distillation to characterize object-level normality. Differently from these previous works, our approach performs multiple predictions instead of a single one for each normality aspect. Furthermore, our method can detect location-dependent anomalies which have not been addressed in the aforementioned methods. To our knowledge, only Doshi & Yilmaz (2020) addressed the modeling of location attributes at the object-level. The authors proposed a non-parametric model of hand-crafted object-level features which included the object position. Differently, we model the distribution of normal bounding boxes separately for each object class which allows to detect class-wise location anomalies.

## 3    Masked multi-prediction for normality modeling: a preliminary study

In this section, we introduce our masked multi-prediction (MMP) approach and motivate it theoretically and experimentally in the context of anomaly detection. We first demonstrate the importance of multi-prediction compared to single prediction. Then, we show the impact of the loss choice to ensure diversity and likelihood of predictions. Finally, we discuss the importance of spatial and temporal masking in improving anomaly detection performance. The proofs of all propositions are provided in the supplementary material. Preliminary experiments to support our analysis have been carried out on the following datasets:

**Synthetic roads dataset:** This dataset is used to show the impact of performing multi-prediction instead of single prediction in the context of anomaly detection. Indeed, we generated a synthetic dataset (Figure 1) illustrating a road scenario in which a car is at the intersection of $m$ roads. In the normal case, the car moves towards one of the roads. More precisely, we assume that the $x$ axis is oriented from left to right and that the $y$ axis is oriented from bottom to top. We indicate the position of the car by its 2D coordinates $(x, y)$. Our synthetic roads dataset consists of sequences of 2 pairs of coordinates indicating the position of the car at times $t$ and $t + 1$. Therefore a sample $X$ is in the form : $X = (p_t, p_{t+1}) = ((x_t, y_t), (x_{t+1}, y_{t+1})) \in [-1, 1]^4$. For simplicity, we will assume that the car is always at position: $p_t = (0, 0)$ at time $t$ and that, in the normal case, it uniformly moves to one of the following positions at time $t + 1$: $\left( p_{t+1}^{(i)} = (1, \frac{2 \times (i-1)}{m-1} - 1) \right)_{i \in [\![1,m]\!]}$.

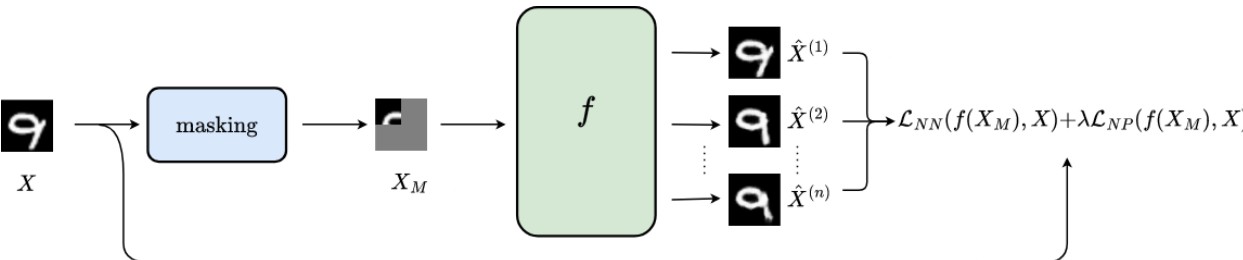

Figure 2: Overview of our methodology: the masked multi-prediction network (MMP) denoted by $f$ receives a masked sample $X_M$ and produces $n$ likely predictions: $f(X_M) = (\hat{X}^{(k)})_{k \in [\![1,n]\!]}$ which are compared to the original sample $X$. The model is trained using $\mathcal{L}_{NN}$ and $\mathcal{L}_{NP}$. At inference only $\mathcal{L}_{NN}$ is used.

The parameter $m \geq 2$ indicates the number of roads the car can take. Therefore, the set of normal samples $\mathcal{N}$ is:

$$\mathcal{N} = \left\{ (p_t, p_{t+1}) | (\exists i \in [\![1, m]\!]) : p_{t+1} = p_{t+1}^{(i)} = \left( 1, \frac{2 \times (i-1)}{m-1} - 1 \right) \right\}$$

The set of anomalies $\mathcal{A}$ is:

$$\mathcal{A} = \left\{ (p_t, p_{t+1}) | (\forall i \in [\![1, m]\!]) : p_{t+1} \neq \left( 1, \frac{2 \times (i-1)}{m-1} - 1 \right) \right\}$$

We assume that the models are trained on $\mathcal{N}$ by predicting the normal positions at time $t+1$: $\left( p_{t+1}^{(i)} \right)_{i \in [\![1,m]\!]}$ given the position at time $t$, and tested on representative samples of $\mathcal{N} \cup \mathcal{A}$, meaning that all normal positions are in the test set and that abnormal positions cover sufficiently $[-1, 1]^2$.

**MNIST** (LeCun et al. (2010)): this dataset is used to show the importance of masking. It contains handwritten digits from 0 to 9. As this dataset was not designed for the one-class setting, it is adapted by considering a particular class as normal in each case. We use the training-testing split of the dataset and perform training only on samples from the class considered normal. During inference, all test data are used. Test samples from the class on which the model has been trained are considered normal, and other classes are considered abnormal.

### 3.1 Multi-prediction vs. single prediction

In the following analysis, we will focus on the general problem of sample prediction (denoted by $X$) given a masked input $X_M = X \odot M$, where $M$ is the mask applied to $X$. The sample prediction problem boils down to learning the conditional probability distribution $\mathbb{Q} \triangleq \mathbb{P}(X|X_M)$, which can be done using a model that receives as input a masked sample $X_M$ and generates candidate samples according to $\mathbb{P}(X|X_M)$. However, learning this distribution via a single prediction model does not capture its multi-modality as shown in Figure 1. Indeed, a single prediction model $g$ is generally trained to predict a sample $X$ by minimizing:

$$\mathcal{L}_{single}(g(X_M), X) \triangleq \|g(X_M) - X\| \tag{1}$$

However, since $X$ is not deterministic given $X_M$, the network minimizes:

$$\bar{\mathcal{L}}_{single}(g(X_M)) \triangleq \mathbb{E}_{X \sim \mathbb{Q}}(\|g(X_M) - X\|) \tag{2}$$

This loss can be interpreted as the average distance across all possible samples $X \sim \mathbb{Q}$. If mean squared error (MSE) is used, the global minimum of the loss is achieved by a model $g^*$ that predicts the conditional expectation of $X$:

**Proposition 1.** *Let $g$ be a single prediction model trained using the $L_2$ norm. The minimum loss is achieved for a model $g^*$ that predicts the conditional expectation:*

$$g^*(X_M) = \mathbb{E}(X|X_M)$$

This shows that the output of a single prediction model is an average of possible samples ($X \sim \mathbb{Q}$) in the best case, which is sub-optimal. In order to better cover normal possibilities, we propose to train a masked multi-prediction model (MMP): $f = (f^{(1)}, f^{(2)}, ..., f^{(n)}) = (f^{(k)})_{k \in [\![1,n]\!]}$ to predict $n$ likely possibilities using the nearest neighbor objective function (Guzmán-rivera et al. (2012)). An illustration of MMP is provided in Figure 2. The nearest neighbor loss penalizes only the distance to the closest prediction. Formally, the loss between the set of predictions $f(X_M) = (f^{(k)}(X_M))_{k \in [\![1,n]\!]}$ and a sample $X$ can be written as follows:

$$\mathcal{L}_{NN}(f(X_M), X) \triangleq \min_{k \in [\![1,n]\!]} \|f^{(k)}(X_M) - X\| \tag{3}$$

The corresponding expected loss across all possible samples $X \sim \mathbb{Q}$ is:

$$\bar{\mathcal{L}}_{NN}(f(X_M)) \triangleq \mathbb{E}_{X \sim \mathbb{Q}}(\mathcal{L}_{NN}(f(X_M), X)) \tag{4}$$

Training a MMP model using the previous loss can achieve a better fitting of normal data (i.e a lower training loss) as shown in the following proposition:

**Proposition 2.** *Let $X$ a sample from $\mathbb{P}$, $\mathcal{F}$ the space of self-maps of $[0, 1]^{C \times H \times W}$ and $f^* \in \arg\min_{f=(f^{(k)})_{k \in [\![1,n]\!]} \in \mathcal{F}^n} \bar{\mathcal{L}}_{NN}(f(X_M))$. The minimum expected loss is lower when using multi-prediction than when using single prediction:*

$$\bar{\mathcal{L}}_{NN}(f^*(X_M)) \leq \bar{\mathcal{L}}_{single}(g^*(X_M))$$

*Moreover, in the case of MSE, by considering the index of the closest predictor to $X$: $K = \arg\min_{k \in [\![1,n]\!]} \|f^{*(k)}(X_M) - X\|_2^2$, we have:*

$$\bar{\mathcal{L}}_{single}(g^*(X_M)) - \bar{\mathcal{L}}_{NN}(f^*(X_M)) = \mathbb{E}_K(\|\mathbb{E}(X|X_M) - \mathbb{E}(X|X_M, K)\|_2^2 | X_M)$$

The last result shows that the prediction loss gap between a single prediction model and an MMP model increases with the distance between optimal single prediction and optimal multiple predictions. Consequently, this gap increases as distribution becomes "more" multi-modal. Moreover, a MMP network trained using the nearest neighbor loss provides a better anomaly detection performance than a single prediction model, as shown in section 5.4 and as demonstrated in the case of the synthetic roads dataset. Indeed, Proposition 3 shows that a MMP network trained using the nearest neighbor objective function $\mathcal{L}_{NN}$ can theoretically achieve a perfect anomaly detection score on the synthetic roads dataset, if it reaches the global minimum of the training loss, whereas a single prediction model cannot, even if the global minimum is reached.

**Proposition 3.** *A MMP model with a number of predictors corresponding to the number of roads $m$ and reaching the global minimum training loss using $\mathcal{L}_{NN}$, achieves an AUC of 100% on the synthetic roads dataset. On the other hand, a single prediction model achieving the global minimum training loss using $\mathcal{L}_{single}$ cannot achieve an AUC of 100%.*

## 3.2 Impact of loss functions

### 3.2.1 Nearest neighbor loss

The choice of the nearest neighbor objective function is important to ensure diversity of predictions and a better fit to normal data (Guzmán-rivera et al. (2012)). Indeed, if we use instead a naive objective function $\mathcal{L}_{naive} \triangleq \frac{1}{n} \sum_{k \in [\![1,n]\!]} \|f^{(k)}(X_M) - X\|$. The diversity of predictions is lost, which amounts to making a single prediction, as shown in the following proposition:

**Proposition 4.** *Let $\bar{\mathcal{L}}_{naive}(f(X_M))$ the expected loss corresponding to $\mathcal{L}_{naive}(f(X_M), X)$ :*

$$\bar{\mathcal{L}}_{naive}(f(X_M)) = \mathbb{E}_{X \sim \mathbb{Q}} \left( \frac{1}{n} \sum_{k \in [\![1,n]\!]} \|f^{(k)}(X_M) - X\| \right) \tag{5}$$

*In case of $L_2$ norm, the minimum expected loss is achieved by a MMP model: $f^* = (f^{*(k)})_{k \in [\![1,n]\!]}$ such that $(\forall k \in [\![1,n]\!]) : f^{*(k)}(X_M) = \mathbb{E}(X|X_M)$, which is similar to perform single prediction.*

### 3.2.2 Non-participation loss

In practice, $\mathbb{P}(X|X_M)$ is often intractable, highly dimensional and we have only access to samples from it (e.g., the future frame observed in a video). Therefore, it is infeasible to train the network via $\bar{\mathcal{L}}_{NN}(f(X_M))$, since it involves an expectation over all possible samples $X$, which is difficult to compute. Instead, we train the network via $\mathcal{L}_{NN}(f(X_M), X)$, where $X$ is the actual observed sample. This loss encourages the model to produce diverse predictions in order to reduce the distance between the sampled data point $X$ and its nearest neighbor $f^{(k^*)}(X_M)$ where $k^* = \arg\min_k \|f^{(k)}(X_M) - X\|$.

Nevertheless, this objective function only optimizes the best among all predictions ($f^{(k^*)}(X_M)$). Thus, the model receives a sparse signal which may lead to optimizing only predictors which are selected as nearest neighbors during training. Other predictors remain far from the data subspace, as they are never selected and therefore never optimized. In order to overcome this issue, we introduce a novel objective function called *the non-participation loss* in order to optimize these predictors. More specifically, we collect the indices of unoptimized predictors $\mathcal{U} \subset [\![1,n]\!]$ that were not selected as nearest neighbors in the last epoch of training (a small threshold $\delta$ is used in practice, cf. Section 5.2). Then, we optimize those predictors : $(f_F^{(p)})_{p \in \mathcal{U}}$ via the non-participation loss that can be written as:

$$\mathcal{L}_{NP}(f(X_M), X) \triangleq \sum_{p \in \mathcal{U}} \|f^{(p)}(X_M) - X\| \tag{6}$$

Thus, we train the network via a weighted combination of the two losses $\mathcal{L} \triangleq \mathcal{L}_{NN} + \lambda \mathcal{L}_{NP}$. The following proposition shows that adding the non-participation objective to the nearest neighbor loss can only decrease the prediction error:

**Proposition 5.** *Let $\mathcal{X}$ the training dataset composed from normal samples. We denote by $\tilde{f} = (\tilde{f}^{(k)})_{k \in [\![1,n]\!]}$ a MMP model optimized via the nearest neighbor objective $\mathcal{L}_{NN}$ and achieving a loss $\mathcal{L}_{NN}(\tilde{f}(X_M), X)$ on a sample $X \in \mathcal{X}$. Let $\hat{f} = (\hat{f}^{(k)})_{k \in [\![1,n]\!]}$ a MMP model resulting from training the non-optimized predictors of $\tilde{f}$ via the non-participation loss $\mathcal{L}_{NP}$, and $\mathcal{L}_{NN}(\hat{f}(X_M), X)$ the loss of $\hat{f}$ on a sample $X$. We have ($\forall X \in \mathcal{X}$):*

$$\mathcal{L}_{NN}(\hat{f}(X_M), X) \leq \mathcal{L}_{NN}(\tilde{f}(X_M), X)$$

Table 1: Influence of training loss choice. The metrics *participation* and *prediction loss* ($\times 10^3$) are calculated as the average obtained by training each model ten times, each time on a different class from MNIST.

| Variant | Losses | Pretraining | $\lambda$ | Participation ↑ | Prediction loss ↓ |
|---|---|---|---|---|---|
| v1 (1 pred.) | $\mathcal{L}_{single} = \mathcal{L}_{NN} = \mathcal{L}_{naive}$ | × | - | 1/1 | 7.04 |
| v2 (10 pred.) | $\mathcal{L}_{naive}$ | × | - | **10/10** | 7.29 |
| v3 (10 pred.) | $\mathcal{L}_{NN}$ | × | - | 1/10 | 7.09 |
| v4 (10 pred.) | | ✓ | - | 4.4/10 | 5.32 |
| v5 (10 pred.) | $\mathcal{L}_{NN} + \lambda \mathcal{L}_{NP}$ | × | 0.01 | **10/10** | 4.57 |
| v6 (10 pred.) | | × | 0.1 | **10/10** | **4.26** |
| v7 (10 pred.) | | × | 1 | **10/10** | 4.42 |
| v8 (10 pred.) | | × | 10 | **10/10** | 4.52 |

In order to empirically verify Propositions 2, 4 and 5, we trained a single prediction model composed of one predictor via $\mathcal{L}_{single}$ as well as three MMP models each composed of $n$ predictors, and each predictor has the same architecture as the single prediction model. The MMP models are trained via the following training objectives: $\mathcal{L}_{NN}$, $\mathcal{L}_{NN} + \lambda\mathcal{L}_{NP}$ and $\mathcal{L}_{naive}$. Since $\mathcal{L}_{NN}$ optimizes only the best among all predictors, we observed that it is very sensitive to the initialisation of the predictors. Thus, we performed two types of training when using this loss, the first consists in randomly initializing predictors (similarly to other losses), and the second consists in pretraining the predictors using $\mathcal{L}_{naive}$. More precisely, we train the predictors using $\mathcal{L}_{naive}$ for the first epoch of training. The results are shown in Table 1. First, we observe that when an MMP model is trained using $\mathcal{L}_{NN}$, it achieves either a lower loss (v4 vs. v1) when the predictors are pretrained, or an equivalent loss in the worst case (v3 vs. v1), when only one predictor is optimized.

Second, we observe that when an MMP network is trained via $\mathcal{L}_{naive}$, it achieves a loss similar to $\mathcal{L}_{single}$, which is in line with proposition 4. Note that the loss is slightly higher when learning an MMP model using $\mathcal{L}_{naive}$ than a single-prediction model, we explain by the fact that the former requires training all predictors instead of just one. When using $\mathcal{L}_{NN} + \lambda\mathcal{L}_{NP}$, all predictors participate in the training, resulting in a lower prediction loss. To test the sensitivity of training to the $\lambda$ parameter, we experimented with different orders of magnitude (v5,v6,v7,v8), and observed slight changes in the prediction loss. One possible explanation for this low sensitivity to the $\lambda$ parameter is that the $\mathcal{L}_{NP}$ loss optimizes predictor parameters that have not been optimized by $\mathcal{L}_{NN}$ (by definition). Consequently, the two losses optimize different sets of parameters since the predictors are independent. In addition, once all predictors start to be selected by $\mathcal{L}_{NN}$, $\mathcal{L}_{NP}$ is no longer used. Additional experiments on baseline models are provided in the supplementary material (section C.1).

### 3.3 Impact of masking

The role of masking is to ensure that the model does not learn the trivial identity function, which would result in a good reconstruction of both normal and abnormal samples, and therefore a poor discrimination between them. Indeed, masking forces the model to learn the specificities of normal data in order to predict well normal samples while having a poor prediction of abnormal samples. This would result in a better detection of anomalies. It is important to note that masking can be applied either spatially or temporally. For example, patch masking, illustrated in Figure 3, is a clear example of spatial masking. On the other hand, in the case of future frame prediction based on a current frame, masking is implicitly performed at the temporal level. In this case, the sample $X$ can be considered as a pair of the current and next frames: $(X_P, X_F)$ and the masked sample $X_M$ is the current frame $X_P$. In order to illustrate the importance of masking, we carried out a comparison between our masked multi-prediction model (MMP) and a reconstruction network, both trained on a single class from MNIST (Figure 3). The training details are provided in the supplementary material. Different masking strategies are presented in Tables 2a and 2b. We observe that when masking is used, the recovery of abnormal patterns is more difficult than that of normal patterns, resulting in better anomaly detection performance (e7 vs. e4 in Table 2b). Moreover, it can be seen that a high percentage of masking is advantageous for anomaly detection on MNIST. However, a compromise arises when selecting the patch size for masks. Specifically, as the patch size increases, the prediction task becomes more challenging, since the model is required to predict a more global information.

Table 2: Influence of masking strategies on anomaly detection performance.

(a) Influence of patch size of masks on anomaly detection performance (Mean AUC) on MNIST, using 75% percentage masking.

| Exp. | Patch size | Mean AUC |
|------|-----------|----------|
| e1 | $8 \times 8$ | 89.0% |
| e2 | $16 \times 16$ | **94.0%** |
| e3 | $32 \times 32$ | 90.2% |

(b) Influence of the percentage of pixels masked on anomaly detection performance (Mean AUC) on MNIST, using a patch size of $16 \times 16$.

| Exp. | Masking percentage | Mean AUC |
|------|-------------------|----------|
| e4 | 0% | 85.9% |
| e5 | 25% | 90.4% |
| e6 | 50% | 91.3% |
| e7 | 75% | **94.0%** |

# 4 Multi-aspect normality modeling

In this section, we present our masked multi-prediction framework for modeling appearance, motion and semantics (MMP-AMS), which is an adaptation of MMP to VAD. An illustration of our framework is presented in Figure 4. First we introduce the notations for this section:

We denote the normality aspects as follows: $F$ for future frame prediction (appearance), $O$ for optical flow prediction (motion), $C$ for class prediction (semantics), and $B_X, B_Y, B_H, B_W$ for the bounding boxes center coordinates, height and width respectively (location). We denote by $A$ one of these aspects which can belong to the set of appearance, motion and semantics aspects: $\Gamma = \{F, O, C\}$ or location aspects: $\Theta = \{B_X, B_Y, B_H, B_W\}$. $f_A$ is the network used for modeling an aspect $A$, if a network is used for multiple aspects we denote it as $f_{\mathcal{A}}$, where $\mathcal{A}$ is the set of modeled aspects. Let $X$ be a normal object extracted from a frame at time $t$, following the normality distribution $\mathbb{P}$ and having a bounding box with center coordinates $(X_{B_X}, X_{B_Y})$, height $X_{B_H}$ and width $X_{B_W}$. The image of $X$ at times $t$ (Present) and $t + 1$ (Future) are denoted respectively by $(X_P, X_F)$. The masked current image of $X$ is denoted by $X_{MP} \triangleq X_P \odot M$, where $M$ is the mask applied to $X_P$ and $\odot$ is the Hadamard product. The one-hot encoding of object classes and the object-level forward optical flow are denoted by $X_C, X_O$ respectively.

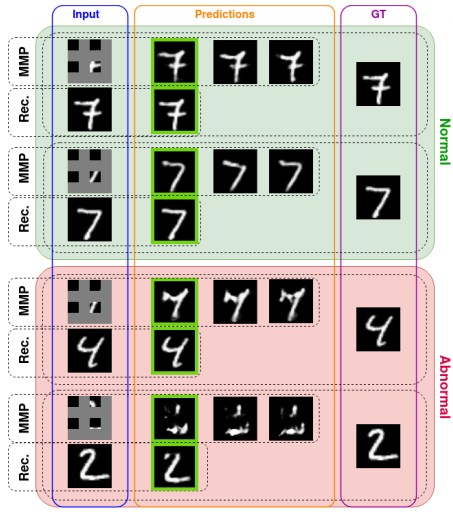

Figure 3: Visualization of two normal samples (class 7) and two abnormal samples (classes 4 and 2) from MNIST. The first column indicates the input of the model, the second column shows the predictions of two models: MMP (ours) which receives a masked input and preforms multiple predictions, and a reconstruction network (Rec.) which receives a non-masked input and preforms a single reconstruction. The nearest predictions are circled in green. The third column indicates the corresponding ground truth.

## 4.1 Overview of MMP-AMS

Our framework learns the appearance, motion and semantics aspects of normality, by predicting multiple future frames as well as the corresponding optical flow and class vectors given a masked current frame. In this way, the model learns: appearance features via unmasking, motion features via future prediction and semantic features through class prediction. For each aspect, the model performs multiple predictions in order to take into account the diversity of normal data w.r.t. to each aspect. We model the three aspects: $\Gamma = \{F, O, C\}$ via three MMP networks $(f_A)_{A \in \Gamma}$ (one for each aspect). Therefore, the full model can be written as: $f_\Gamma \triangleq (f_A)_{A \in \Gamma} = \left( (f_A^{(k)})_{k \in [\![1,n]\!]} \right)_{A \in \Gamma}$. The model predicts respectively different possible future frames $(\hat{X}_F^{(k)})_{k \in [\![1,n]\!]}$, optical flows $(\hat{X}_O^{(k)})_{k \in [\![1,n]\!]}$ and classes $(\hat{X}_C^{(k)})_{k \in [\![1,n]\!]}$ given a masked current frame $X_{MP}$. The $k$-th prediction for an aspect $A \in \Gamma$ is given by: $\hat{X}_A^{(k)} \triangleq f_A^{(k)}(X_{MP})$. The full model $f_\Gamma$ is trained to minimize the nearest neighbor loss and the non-participation loss for each of the three aspects. Specifically, the following sum is minimized:

$$\mathcal{L}_\Gamma(f_\Gamma(X_{MP}), X) \triangleq \sum_{A \in \Gamma} \mathcal{L}_{NN}(f_A(X_{MP}), X_A) + \lambda \mathcal{L}_{NP}(f_A(X_{MP}), X_A) \tag{7}$$

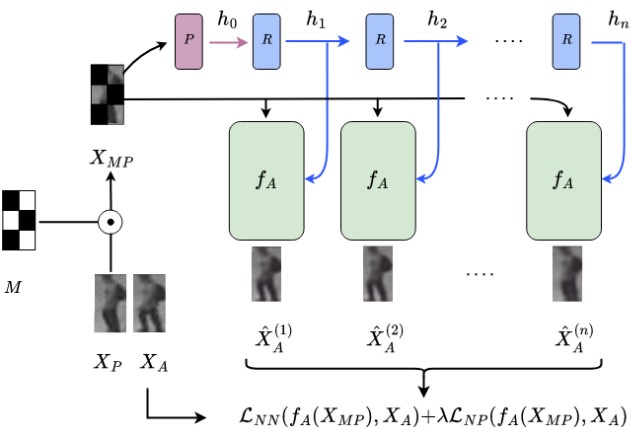

Figure 4: Overview of the proposed masked multi-prediction framework applied to appearance, motion and semantics (MMP-AMS). We show the model predictions for an aspect $A$, which can be one of the following aspects: future frame prediction $F$ (appearance), optical flow prediction $O$ (motion) or class prediction $C$ (semantics). We consider $A = F$ in the illustration. The network receives a masked image at time $t$ and produces multiple predictions $(\hat{X}_A^{(k)})_{k \in [\![1,n]\!]}$ for an aspect $A$. The model is trained using $\mathcal{L}_{NN}$ and $\mathcal{L}_{NP}$. At inference only $\mathcal{L}_{NN}$ is used. Networks of the same color share parameters.

## 4.2 Network design

In order to allow the number of predictions to be varied without changing the total number of model parameters, we introduce a novel architecture which consists of a recurrent neural network $R$, a first state predictor $P$ as well as three state conditional networks $(f_A)_{A \in \Gamma} = \left( (f_A(X_{MP}, h_k))_{k \in [\![1,n]\!]} \right)_{A \in \Gamma}$ (illustrated in Figure 4). For each aspect $A$, we propose to replace the predictors $(f_A^{(k)})_{k \in [\![1,n]\!]}$ by a state conditional network $f_A$ which receives as input the masked frame $X_{MP}$ as well as a state $h_k$ generated using the recurrent neural network $R$. Therefore, $k$-th predictor $f_A^{(k)}$ is equivalent to conditioning the network $f_A$ on the state $h_k$. Formally, we have $(\forall A \in \{F, O, C\}) : \hat{X}_A^{(k)} = f_A^{(k)}(X_{MP}) = f_A(X_{MP}, h_k) = f_A(X_{MP}, R(h_{k-1}))$. The recurrent architecture ensures that the hidden state $h_k$ contains information about previously predicted states, and encourages the model to explore new normality regions in order to minimize the nearest neighbor error. The networks architectures, the number of parameters and training pseudo-code are provided in the supplementary material.

## 4.3 Location module

In order to detect location related anomalies, we learn the distribution of object positions for each normal class using a simple Gaussian model. More specifically, given a class $X_C$, we model the distribution of bounding box centers $(X_{B_X}, X_{B_Y})$, height and width $(X_{B_H}, X_{B_W})$ of objects belonging to $X_C$, using 4 Gaussians: $\mathcal{N}(\alpha_A(X_C), \beta_A(X_C))$ for $A \in \Theta = \{B_X, B_Y, B_H, B_W\}$. The mean $\alpha_A(X_C)$ and the standard deviation $\beta_A(X_C)$ for a given dimension $A$ are predicted by a network $f_\Theta$ using the negative log-likelihood loss:

$$\mathcal{L}_\Theta(f_\Theta(X_C), X) \triangleq \frac{1}{2} \sum_{A \in \Theta} \log(\beta_A(X_C)) + \left( \frac{X_A - \alpha_A(X_C)}{\beta_A(X_C)} \right)^2 \tag{8}$$

### 4.4 Anomaly scoring

At inference time, we perform the same pre-processing steps on a test sample and compute the appearance, motion and semantics anomaly scores $S_\Gamma(X)$ as well as the location anomaly scores $S_\Theta(X)$ by summing the $z$-scores of the prediction errors across aspects. This allows balancing the contribution of each aspect to the anomaly score. More formally, given an object $X$ at frame $\mathcal{F}_t$:

$$S_\Gamma(X) \triangleq \sum_{A \in \Gamma} w_A \frac{\mathcal{L}_{NN}(f_A(X_{MP}), X_A) - \mu_A}{\sigma_A} \quad (9) \quad S_\Theta(X) \triangleq \sum_{A \in \Theta} w_A \frac{\mathcal{L}_{NLL}(X_A; (\alpha_A, \beta_A)) - \mu_A}{\sigma_A} \quad (10)$$

where $f_A, \alpha_A, \beta_A$ are the networks and parameters after training. $\mu_A$ and $\sigma_A$ are respectively the expectation and the standard deviation of the loss function which is either $\mathcal{L}_{NN}$ or $\mathcal{L}_{NLL}$ estimated from normal training data for a given aspect $A$. $w_A$ is the weight assigned to each aspect, which may vary depending on the application. The final anomaly score is a weighted combination of the appearance, motion, semantics and location anomaly scores: $S(X) \triangleq S_\Gamma(X) + \gamma S_\Theta(X)$. The parameter $\gamma$ can be chosen depending on whether we aim to detect location anomalies. The frame level score for a frame is the maximum object-level score in the frame.

## 5 Experimental study

### 5.1 Datasets and evaluation metrics

We adopt the following metrics: the Region-Based Detection Criterion (RBDC) and the Track-Based Detection Criterion (TBDC) introduced by Ramachandra & Jones (2020) as an alternative to frame-level AUC widely used in the literature. The latter metric measures the anomaly detection performance at the temporal level only. However, it does not evaluate the capacity of the model to localize anomalies spatially because it does not penalize false positive regions detected in abnormal frames as pointed out by Ramachandra & Jones (2020). We perform experiments on the most commonly used datasets for the one-class and object-centric scenario. **UCSDped2** (Mahadevan et al. (2010)) is a single scene dataset which includes anomalies such as riding a bike and driving a vehicle on a sidewalk. Ramachandra & Jones (2020) provided region-level and track-level annotations for the RBDC and TBDC metrics. **ShanghaiTech** (Luo et al. (2017)) contains scenes of different backgrounds. Anomalies include jumping, running, or stalking on a sidewalk. The region-level and track-level annotations are provided Georgescu et al. (2021b). **CUHK Avenue** (Lu et al. (2013)) is a single scene dataset which consists of videos with abnormal events such as running or walking towards the camera. We use the improved set of annotations proposed by Ramachandra & Jones (2020) which take into account some static anomalies that where not considered in the original annotations.

### 5.2 Implementation details

For a fair comparison to other object-centric approaches, Yolov3 (Redmon & Farhadi (2018)) pretrained on MSCOCO is applied for object detection, using the implementation of MMDetection (Chen et al. (2019)) with an objectness threshold of 0.5 for UCSDped2 since objects have low resolutions and 0.7 for Avenue and ShanghaiTech. The set of objects detected by the used implementation contained very small false positives which are filtered out based on their area (lower than 350 pixels). Optical flow maps are computed using the official implementation of FlowNet2 (Reda et al. (2017)) as in (Liu et al. (2021)). For anomaly scoring, we keep only the optical flow magnitudes since the optical flow orientation maps are not precise enough to be predicted for small displacements. The detected objects as well as the corresponding optical flow maps are resized to 64x64. For the mask $M$, we remove 50% of pixels using a grid of 4x4 pixels. Regarding the distance used for anomaly scoring in Section 4, we use the $L_1$ distance for RGB and optical flow, as well as the cross entropy loss for class probabilities. We train the network for 150 epochs for UCSDped2 and Avenue and for 400 epochs for ShanghaiTech using Adam optimizer with a learning rate of $10^{-3}$ with a batch size of 640 for the biggest dataset ShanghaiTech and 64 for UCSDped2 and Avenue. We set $(\gamma, w_I, w_F, w_C, w_{B_X}, w_{B_Y}, w_{B_H}, w_{B_W}, \lambda) = (0, 1, 1, 1, 1, 1, 1, 1, 0.1)$ for UCSDped2 and ShanghaiTech

Table 3: Comparison of our approach to state-of-the-art object-centric VAD methods on RBDC and TBDC (%). Best results are in bold, second best are underlined.

| Method | UCSDped2 | | ShanghaiTech | | Avenue | |
|---|---|---|---|---|---|---|
| | RBDC | TBDC | RBDC | TBDC | RBDC | TBDC |
| Ionescu et al. (2019a) | 52.8 | 72.9 | 20.7 | 44.5 | 15.8 | 27.0 |
| Liu et al. (2021) | - | - | - | - | 41.1 | 86.2 |
| Georgescu et al. (2021b) | 69.2 | 93.2 | 41.3 | 78.8 | 65.1 | 66.9 |
| Georgescu et al. (2021a) | 72.8 | 91.2 | 42.8 | 83.9 | 57.0 | 58.3 |
| Bergaoui et al. (2022) | 80.1 | 95.4 | 51.5 | 82.2 | 75.8 | 70.0 |
| Naji et al. (2022) | 77.2 | 98.5 | 51.6 | 84.6 | 75.3 | 73.4 |
| Georgescu et al. (2021a) + Ristea et al. (2022) | - | - | 40.6 | 83.5 | 66.0 | 64.9 |
| Liu et al. (2018) + Ristea et al. (2022) | - | - | 18.5 | 60.2 | 20.1 | 62.3 |
| Liu et al. (2021) + Ristea et al. (2022) | - | - | 45.5 | 84.5 | 62.3 | **89.3** |
| Barbalau et al. (2022) | - | - | 47.1 | 85.6 | 47.8 | 85.2 |
| Ours (MMP-AMS w/o location module) | **84.0** | **99.0** | **55.9** | **85.7** | 67.4 | 68.0 |
| Ours (MMP-AMS w/ location module) | | | | | **77.7** | 74.2 |

and $(1, 1, 0.1, 0.1, 1, 1, 1, 1, 0.1)$ for Avenue. We explain the parameters choices in Section 5.4. For the non-participation loss, we select predictors which have a participation below $\delta = 5\%$. Regarding the inference time, our model processes a batch of objects in a frame taken from Avenue in 18ms on a single Nvidia-Titan-X GPU. Therefore, it satisfies the real-time constraints, given real-time object detector and optical flow extractor that can run in parallel. In our implementation, the inference time of Yolov3 is 50 ms and that of FlowNet2 is 55 ms. Therefore, if optical flow estimation and object detection are parallelized, our pipeline can run at 13 FPS. Furthermore, the method requires access to one future frame only to compute the anomaly score which allows online application.

## 5.3 Evaluation results

This section presents the results (Table 3) of our method on the benchmarks UCSDped2, Avenue, ShanghaiTech. Since we focus on object-level anomalies and for a fair comparison with state-of-the-art object-centric methods, we use the same baseline object-detector (Yolov3). Qualitative results are provided in the supplementary material.

**UCSDped2.** On this dataset, MMP-AMS outperforms previous works on RBDC (+3.9 p.p.), and reaches the state-of-the-art on TBDC. It can be seen that the optical flow prediction is particularly relevant for this dataset (cf. ablation study in Section 5.4) due to the fact that most anomalies have abnormal motion.

**ShanghaiTech.** Unlike other datasets, this one contains multiple scenes for training and testing. Nevertheless it shares a similar normal context across scenes. Since our method is object-centric, it is less sensitive to scene changes. MMP-AMS achieves a significant improvement (+4.3 p.p.) in terms of RBDC and slightly outperforms other methods on TBDC. The improvements in terms of anomaly localization can be explained by two factors. First, the choice of aspects allows us to detect appearance, semantics and behavior anomalies (cf. ablation study in Section 5.4). Second, we found that multi-prediction is beneficial for this dataset, which is explained by a high scene complexity leading to a multiplicity of normal scenarios.

**Avenue.** This dataset is challenging because it contains several types of anomalies, such as human behavior and unusual objects. In addition, unlike previous datasets, it contains location-dependent anomalies. It is important to note that no method consistently outperforms the others in all metrics. However, MMP-AMS combined with the location module provides a good compromise between RBDC and TBDC. More specifically, our approach achieves the best performance in terms of RBDC (+1.9 p.p.) and a moderate TBDC. Nevertheless, our method achieves the best TBDC (+0.8 p.p.) among methods which use a similar temporal window (Georgescu et al. (2021b); Bergaoui et al. (2022); Naji et al. (2022)). A more detailed analysis of the impact of increasing the temporal window is provided in the supplementary material. The

Table 4: Comparison between different training loss functions using the metrics *participation*, *diversity* and *prediction loss* computed for normal samples from Avenue using the RGB modality. The *participation* is the selection frequency of a predictor. The *diversity* ($\times 10^2$) is the average pixelwise distance between predictions from two different predictors (higher is better). The *prediction loss* is the nearest neighbor loss (lower is better). Green and red respectively indicate best and worst results.

| Variant | Losses | Training metrics | | |
|---|---|---|---|---|
| | | Participation | Diversity ↑ | Prediction loss ↓ |
| v1 (1 pred.) | $\mathcal{L}_{naive} = \mathcal{L}_{NN}$ | 100% | 0 | 0.164 |
| v2 (3 pred.) | $\mathcal{L}_{naive}$ | 29%, 13%, 58% | 0.03 | 0.165 |
| v3 (3 pred.) | $\mathcal{L}_{NN}$ | 65%, 35%, 0% | 1.4 | 0.159 |
| v4 (3 pred.) | $\mathcal{L}_{NN} + \lambda\mathcal{L}_{NP}$ | 35%, 51%, 14% | 1.8 | 0.157 |

good performance in RBDC is partly due to simultaneously taking into account the appearance, motion and location aspects which are relevant for this dataset. Moreover, MMP-AMS alone outperforms methods designed for appearance and motion anomalies (Georgescu et al. (2021b;a); Barbalau et al. (2022); Liu et al. (2021); Ristea et al. (2022); Ionescu et al. (2019a)) in terms of RBDC. When combined with the location module, it outperforms all methods under consideration. This shows the complementarity of our two modules for this dataset.

## 5.4 Discussion

### 5.4.1 Impact of multi-prediction

As mentioned in the introduction, there are trade-offs between reconstruction-based methods and future prediction-based methods. While the former reconstruct training data well, they also tend to reconstruct anomalies. On the contrary, the latter predict anomalies poorly, however, they predict less well normal data. Our approach embraces advantages of both families. Indeed, as our model has only access to a masked current image, it cannot recover anomalies well, and thanks to multi-prediction, it fits normal data better than single-prediction methods, as shown in Proposition 1. In order to empirically verify our claims, we compared MMP-AMS performing only one prediction (v1 in Table 4) with the same framework performing multiple predictions (v2,v3,v4 in Table 4). We observe in Table 4 that a multi-prediction network achieves a lower prediction error than a single prediction network only when $\mathcal{L}_{NN}$ is used ((v2,v3,v4) vs. v1). On the one hand, we observe that the model actually produces similar predictions when trained with $\mathcal{L}_{naive}$ (v2 in Table 4), which is coherent with Proposition 4. On the other hand, when we train the model using $\mathcal{L}_{NN}$, it produces a higher diversity of predictions (v3, v4 in Table 4).

Indeed, $\mathcal{L}_{NN}$ encourages predictor specialization, since it penalizes only the best guess. This observation is consistent with Proposition 1. Nevertheless, optimizing $\mathcal{L}_{NN}$ alone leads to the non-participation (v3 in Table 4) of some predictors in the training, since they are never selected as nearest neighbors. This explains the introduction of the non-participation loss $\mathcal{L}_{NP}$ (v4 in Table 4) which ensures that all branches get optimized. Empirically, we notice that it allows the model to better fit normal data since it helps to decrease the prediction loss and increases diversity. In terms of anomaly detection performance, we trained the MMP-AMS framework to predict up to 4 predictions on Avenue (Figure 5). We can observe a significant increase in

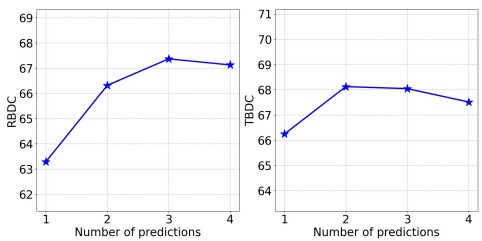

Figure 5: Influence of the number of predictions on MMP-AMS performance (RBDC, TBDC % scores on Avenue).

Table 5: RBDC and TBDC scores in % obtained by incrementally combining the aspects. Best performances are in bold. The normality aspects are denoted as follows: *F* for future frame prediction, *O* for optical flow prediction, *C* for class prediction.

| Ablation | Location module | Set of aspects | UCSDped2 | | ShanghaiTech | | Avenue | |
|---|---|---|---|---|---|---|---|---|
| | | | RBDC | TBDC | RBDC | TBDC | RBDC | TBDC |
| a1 | $\times$ | $\{C\}$ | 64.7 | 75.2 | 37.5 | 58.8 | 04.5 | 18.0 |
| a2 | $\times$ | $\{O\}$ | 82.5 | **99.5** | 48.7 | 84.2 | 17.2 | 54.5 |
| a3 | $\times$ | $\{F_{\text{w/o masking}}\}$ | 61.9 | 82.1 | 47.2 | 82.2 | 67.5 | 66.7 |
| a4 | $\times$ | $\{F\}$ | 66.8 | 85.1 | 48.4 | 83.4 | 68.2 | 66.7 |
| a5 | $\times$ | $\{F, C\}$ | 76.7 | 94.9 | 53.8 | 83.2 | 68.5 | 67.5 |
| a6 | $\times$ | $\{F, C, O\}$ | **84.0** | 99.0 | **55.9** | **85.7** | 67.4 | 68.0 |
| a7 | $\checkmark$ | $\{F, C, O\}$ | 80.5 | 94.2 | 52.6 | 83.1 | **77.7** | **74.2** |

Table 6: Influence of masking strategies on anomaly detection performance on UCSDped2.

(a) Influence of patch size of masks on anomaly detection performance, using 50% percentage masking.

(b) Influence of the percentage of pixels masked on anomaly detection performance, using a patch size of $4 \times 4$.

| Exp. | Patch size | RBDC | TBDC |
|---|---|---|---|
| e1 | $4 \times 4$ | 66.8 | **85.1** |
| e2 | $8 \times 8$ | **67.6** | 82.3 |
| e3 | $16 \times 16$ | 65.3 | 79.2 |

| Exp. | Masking percentage | RBDC | TBDC |
|---|---|---|---|
| e4 | 0% | 61.9 | 82.1 |
| e5 | 25% | 66.6 | 82.3 |
| e6 | 50% | **66.8** | **85.1** |
| e7 | 75% | 61.4 | 79.8 |

all metrics (RBDC: +4.1 p.p., TBDC: +1.8 p.p.) until 3 predictions. However, in the case of 4 predictions, performance decreases slightly but remains superior to that of a single prediction. This suggests that 3 predictions are enough to model the diversity of normality for this dataset.

### 5.4.2 Impact of the choice of normality aspects

In MMP-AMS, we introduced multiple aspects to capture diverse types of normality patterns. This allows the model to jointly learn spatio-temporal fine grained patterns via unmasking and future prediction, as well as the object-level semantics through class prediction. Those aspects are complementary, especially for datasets that contain diverse appearance, motion and semantics anomaly types such as ShanghaiTech. This results in performance improvement when incrementally adding more aspects in the normality modeling (a2, a3, a4 in Table 5). We notice that class and optical flow predictions are less relevant for Avenue dataset. This can be explained by two reasons: 1) most anomalies on this dataset are done by humans for which the class information is not relevant to detect anomalies; 2) optical flow predictions are not enough to characterize complex motion patterns in the scene that would require additional 3D information. Therefore, we give them less weight for anomaly scoring (cf. weights $w_A$ detailed in Section 5.2). Regarding the masking of the input (cf. a2 vs. a1 in Table 5), we can observe that it is beneficial for all datasets especially for UCSDped2. This suggests that constraining the prediction task by masking the input makes the prediction even harder for abnormal objects, which leads to a better discrimination between normal and abnormal samples. The influence of masking parameters is presented in Tables 8a and 8b. We observe that optimal performances are achieved using less masking compared to MNIST, which can be explained by the higher complexity of the urban scenes compared to MNIST images. Concerning the location module, modeling the distribution of bounding boxes (cf. a6 vs. a7 Table 5) significantly improves the results for Avenue dataset. This can be explained by the fact that some anomalies in Avenue are related to the position with respect to the camera, which is not the case for UCSDped2 and ShanghaiTech, that do not include location-dependent anomalies in the definition of what is considered as abnormal. For the sake of consistency with the definition of anomalies in these datasets, the location aspect is not taken into account (cf. parameter $\gamma = 0$ as detailed in Section

5.2). We explain the performance drop when adding the location module to MMP-AMS (cf. a6 vs. a7 Table 5) for those datasets, by the fact that the full model (MMP-AMS w/ location module) becomes able to detect location related anomalies, however, those types of anomalies are not included in the definition of what is abnormal in both UCSDped2 and ShanghaiTech. This means that samples, for which **only** the location aspect is abnormal (cf. Figure 3 in the supplementary material), are marked as normal by the ground truth for those datasets, but since the location module detects those samples as abnormal (since they have an abnormal location), they are considered as an anomaly by the whole model (MMP-AMS w/ location module), therefore leading to a false positive detection w.r.t. the ground truth for those datasets. Thus, even if the location module detects location related anomalies well, we still can observe a drop in performance w.r.t. the ground truth for those datasets. These results show the importance of defining the aspects of normality that are relevant to each user application, in order to achieve optimal anomaly detection performance. As our approach models these aspects via separate networks, it allows aspects of normality to be weighted according to their relevance to the types of anomalies to be detected. This also provides an end-user explanation of which aspects cause an anomaly score that would trigger an alarm.

### 5.4.3 Limitations and Future work

One downside of our approach is that it depends on supervised object detectors which are usually trained in a closed world manner. However, for some applications, anomaly detection can be aimed at finding objects not seen during training. To address this, it would be interesting to expand our method using open-set object detectors, which are better at adapting to out-of-distribution objects. This could reduce the number of missed anomalies caused by missed detection. Moreover, our method can be improved by adding further normality aspects, which are useful for a given application. For example, it would be interesting to model long-term dependencies such as trajectories. This improvement is valuable for spotting anomalies like loitering, where having a longer context is crucial for an accurate identification.

### 5.4.4 Broader impact

Our work respects the ethical guidelines applied in contemporary computer vision research. Although anomaly detection models can advantageously address user's needs in many applications (e.g. for safety, security and efficiency in autonomous vehicles, video-surveillance, industrial inspection, etc), some negative impacts could arise in the context of misuse or unintended uses, as any machine learning model. Therefore, it is very important to take ethical considerations such as safety, fairness and privacy into account when deploying these models.

## 6 Conclusion

In this work, we addressed the problem of modeling a heterogeneous and multi-aspect normality. For this purpose, we proposed a masked multiple prediction approach (MMP) that is adapted to the multiplicity of possible scenarios. We showed both theoretically and experimentally that modeling the distribution of normal data via multiple predictions improves normality learning and anomaly detection performance. We also discussed the importance of determining the relevant aspects of normality for a given application in order to achieve satisfactory performance, and proposed to model several important aspects of normality such as appearance, motion, semantics and localization. As we model each aspect separately, our approach has the advantage of being both interpretable and modular.

## Acknowledgement

This publication was made possible by the use of the Factory-AI supercomputer, financially supported by the Ile-de-France Regional Council.

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

## A   Propositions and proofs:

**Proposition 1.** *Let $g$ be a single prediction model trained using the $L_2$ norm. The minimum loss is achieved for a model $g^*$ that predicts the conditional expectation:*

$$g^*(X_M) = \mathbb{E}(X|X_M)$$

*Proof.* By definition, the conditional expectation is the minimizer of MSE, we have $(\forall g \in \mathcal{F})$ :

$$
\begin{aligned}
&\bar{\mathcal{L}}_{single}(\mathbb{E}(X|X_M)) \\
&= \mathbb{E}_{X\sim\mathbb{Q}}(\|\mathbb{E}(X|X_M) - X\|) \\
&\leq \mathbb{E}_{X\sim\mathbb{Q}}(\|g(X_M) - X\|) \quad \text{(by definition)} \\
&= \bar{\mathcal{L}}_{single}(g(X_M))
\end{aligned}
$$

Thus, the minimim is reached for:

$$g^*(X_M) = \mathbb{E}(X|X_M)$$

$\square$

**Proposition 2.** *Let $X$ a sample from $\mathbb{P}$, $\mathcal{F}$ the space of self-maps of $[0,1]^{C\times H\times W}$ and $f^* \in \arg\min_{f=(f^{(k)})_{k\in[\![1,n]\!]}\in\mathcal{F}^n} \bar{\mathcal{L}}_{NN}(f(X_M))$. The minimum expected loss is lower when using multi-prediction than when using single prediction:*

$$\bar{\mathcal{L}}_{NN}(f^*(X_M)) \leq \bar{\mathcal{L}}_{single}(g^*(X_M))$$

*Moreover, in the case of MSE, by considering the index of the closest predictor to $X$: $K = \arg\min_{k\in[\![1,n]\!]} \|f^{*(k)}(X_M) - X\|_2^2$, we have:*

$$\bar{\mathcal{L}}_{single}(g^*(X_M)) - \bar{\mathcal{L}}_{NN}(f^*(X_M)) = \mathbb{E}_K(\|\mathbb{E}(X|X_M) - \mathbb{E}(X|X_M, K)\|_2^2 | X_M)$$

*Proof.* **First we show that:** $\bar{\mathcal{L}}_{NN}(f^*(X_M)) \leq \bar{\mathcal{L}}_{single}(g^*(X_M))$

The multi-prediction optimization problem using the nearest neighbor loss can be formulated as follows:

$$\underset{f=(f^{(k)})_{k\in[\![1,n]\!]}\in\mathcal{F}^n}{\text{minimize}} \quad \bar{\mathcal{L}}_{NN}(f(X_M))$$

Where $f$ is multi-prediction model which maps the input $X_M$ to $n$ predictions : $(f^{(k)}(X_M))_{k\in[\![1,n]\!]}$. In particular, if we consider a MMP model for which all predictors produce an optimal single prediction: $g^{(1)}(X_M) = g^{(2)}(X_M) = ... = g^{(n)}(X_M) = g^*(X_M)$, the loss for this particular model is:

$$
\begin{aligned}
\bar{\mathcal{L}}_{NN}(g(X_M)) &= \mathbb{E}_{X\sim\mathbb{Q}}\left(\min_{k\in[\![1,n]\!]}\|g^{(k)}(X_M) - X\|\right) \\
&= \mathbb{E}_{X\sim\mathbb{Q}}\left(\min_{k\in[\![1,n]\!]}\|g^*(X_M) - X\|\right) \\
&= \mathbb{E}_{X\sim\mathbb{Q}}\left(\|g^*(X_M) - X\|\right) \\
&= \bar{\mathcal{L}}_{single}(g^*(X_M))
\end{aligned}
$$

By definition, $\bar{\mathcal{L}}_{NN}(f^*(X_M))$ is the minimum value of the loss $\bar{\mathcal{L}}_{NN}(f(X_M))$. Thus, we have ($\forall f \in \mathcal{F}^n$):

$$\bar{\mathcal{L}}_{NN}(f^*(X_M)) \leq \bar{\mathcal{L}}_{NN}(f(X_M))$$

In particular for $g$:

$$\bar{\mathcal{L}}_{NN}(f^*(X_M)) \leq \bar{\mathcal{L}}_{NN}(g(X_M)) = \bar{\mathcal{L}}_{single}(g^*(X_M))$$

**Second, we show that:** $\bar{\mathcal{L}}_{single}(g^*(X_M)) - \bar{\mathcal{L}}_{NN}(f^*(X_M)) = \mathbb{E}_K(\|\mathbb{E}(X|X_M) - \mathbb{E}(X|X_M, K)\|_2^2 | X_M)$

Let $f^* \in \underset{f=(f^{(k)})_{k\in[\![1,n]\!]}\in\mathcal{F}^n}{\arg\min} \bar{\mathcal{L}}_{NN}(f(X_M))$, and let $K$ defined as: $K = \arg\min_{k\in[\![1,n]\!]} \|f^{*(k)}(X_M) - X\|_2^2$.

We first show that:

$$\bar{\mathcal{L}}_{NN}(f^*(X_M)) = \mathbb{E}(\min_k \|\mathbb{E}(X|X_M, K = k) - X\|_2^2 | X_M)$$

We have:

$$
\begin{aligned}
\bar{\mathcal{L}}_{NN}(f^*(X_M)) &= \mathbb{E}(\min_k \|f^{*(k)}(X_M) - X\|_2^2) \\
&= \mathbb{E}(\min_k \|f^{*(k)}(X_M) - X\|_2^2 | X_M) \\
&= \mathbb{E}(\mathbb{E}(\min_k \|f^{*(k)}(X_M) - X\|_2^2 | K) | X_M) \\
&= \mathbb{E}(\mathbb{E}(\|f^{*(K)}(X_M) - X\|_2^2 | K) | X_M) \\
&= \mathbb{E}(\mathbb{E}(\|f^{*(K)}(X_M) - \mathbb{E}(X|X_M, K) + \mathbb{E}(X|X_M, K) - X\|_2^2 | K) | X_M) \\
&= \mathbb{E}(\mathbb{E}(\|f^{*(K)}(X_M) - \mathbb{E}(X|X_M, K)\|_2^2 + \|\mathbb{E}(X|X_M, K) - X\|_2^2 \\
&\quad + 2 \times (f^{*(K)}(X_M) - \mathbb{E}(X|X_M, K))^T(\mathbb{E}(X|X_M, K) - X)|K)|X_M) \\
&= \mathbb{E}(\mathbb{E}(\|f^{*(K)}(X_M) - \mathbb{E}(X|X_M, K)\|_2^2 | K) | X_M) + \mathbb{E}(\mathbb{E}(\|\mathbb{E}(X|X_M, K) - X\|_2^2 | K) | X_M) \\
&\quad + 2 \times \underbrace{\mathbb{E}(\mathbb{E}((f^{*(K)}(X_M) - \mathbb{E}(X|X_M, K))^T(\mathbb{E}(X|X_M, K) - X)|K)|X_M)}_{=0} \\
&= \mathbb{E}_K(\|f^{*(K)}(X_M) - \mathbb{E}(X|X_M, K)\|_2^2 | X_M) + \mathbb{E}(\mathbb{E}(\|\mathbb{E}(X|X_M, K) - X\|_2^2 | K) | X_M) \\
&= \mathbb{E}_K(\|f^{*(K)}(X_M) - \mathbb{E}(X|X_M, K)\|_2^2 | X_M) + \mathbb{E}(\mathbb{E}(\min_k \|\mathbb{E}(X|X_M, K = k) - X\|_2^2 | K) | X_M) \\
&= \underbrace{\mathbb{E}_K(\|f^{*(K)}(X_M) - \mathbb{E}(X|X_M, K)\|_2^2 | X_M)}_{\geq 0} + \underbrace{\mathbb{E}(\min_k \|\mathbb{E}(X|X_M, K = k) - X\|_2^2 | X_M)}_{\geq \bar{\mathcal{L}}_{NN}(f^*(X_M)) \text{(by definition)}}
\end{aligned}
$$

Thus:

$$\bar{\mathcal{L}}_{NN}(f^*(X_M)) = \mathbb{E}(\min_k \|\mathbb{E}(X|X_M, K = k) - X\|_2^2 | X_M)$$

We will show that $\bar{\mathcal{L}}_{single}(g^*(X_M)) - \bar{\mathcal{L}}_{NN}(f^*(X_M)) = \mathbb{E}_K(\|\mathbb{E}(X|X_M) - \mathbb{E}(X|X_M, K)\|_2^2 | X_M)$:

$$
\begin{aligned}
\bar{\mathcal{L}}_{single}(g^*(X_M)) &= \mathbb{E}(\|\mathbb{E}(X|X_M) - X\|_2^2) \quad \text{(by definition)} \\
&= \mathbb{E}(\|\mathbb{E}(X|X_M) - X\|_2^2|X_M) \\
&= \mathbb{E}(\mathbb{E}(\|\mathbb{E}(X|X_M) - X\|_2^2|K)|X_M) \\
&= \mathbb{E}(\mathbb{E}(\|\mathbb{E}(X|X_M) - \mathbb{E}(X|X_M,K) + \mathbb{E}(X|X_M,K) - X\|_2^2|K)|X_M) \\
&= \mathbb{E}(\mathbb{E}(\|\mathbb{E}(X|X_M) - \mathbb{E}(X|X_M,K)\|_2^2 + \|\mathbb{E}(X|X_M,K) - X\|_2^2 \\
&\quad + 2 \times (\mathbb{E}(X|X_M) - \mathbb{E}(X|X_M,K))^T(\mathbb{E}(X|X_M,K) - X)|K)|X_M) \\
&= \mathbb{E}(\mathbb{E}(\|\mathbb{E}(X|X_M) - \mathbb{E}(X|X_M,K)\|_2^2|K)|X_M) + \mathbb{E}(\mathbb{E}(\|\mathbb{E}(X|X_M,K) - X\|_2^2|K)|X_M) \\
&\quad + 2 \times \underbrace{\mathbb{E}(\mathbb{E}((\mathbb{E}(X|X_M) - \mathbb{E}(X|X_M,K))^T(\mathbb{E}(X|X_M,K) - X)|K)|X_M)}_{=0} \\
&= \mathbb{E}_K(\|\mathbb{E}(X|X_M) - \mathbb{E}(X|X_M,K)\|_2^2|X_M) + \mathbb{E}(\min_k\|\mathbb{E}(X|X_M,K=k) - X\|_2^2|X_M) \\
&= \mathbb{E}_K(\|\mathbb{E}(X|X_M) - \mathbb{E}(X|X_M,K)\|_2^2|X_M) + \bar{\mathcal{L}}_{NN}(f^*(X_M))
\end{aligned}
$$

$\square$

**Proposition 3.** *A MMP model with a number of predictors corresponding to the number of roads $m$ and reaching the global minimum training loss using $\mathcal{L}_{NN}$, achieves an AUC of 100% on the synthetic dataset. On the other hand, a single prediction model achieving the global minimum training loss using $\mathcal{L}_{single}$ cannot achieve an AUC of 100%.*

Before moving on to the demonstration, we need to prove the following lemma:

*Let $\mathcal{N}$ a discrete and finite set of normal samples, $\mathcal{A}$ a set of abnormal samples and $S$ the anomaly scores given by a trained model. We have the following equivalence:*

$$
((\forall(p_n, p_a) \in \mathcal{N} \times \mathcal{A}) : S(p_n) < S(p_a)) \Leftrightarrow AUC = 100\%
$$

*Proof.* $AUC = 100\%$ is equivalent to find a threshold for the anomaly scores that separates perfectly normal and abnormal data. Which is equivalent to having:

$$
(\exists\theta \in \mathbb{R}_+)(\forall(p_n, p_a) \in \mathcal{N} \times \mathcal{A}) : S(p_n) \leq \theta < S(p_a) \tag{11}
$$

Indeed, if $\theta$ exits, the ROC curve passes through the point $(FPR, TPR) = (0, 1)$ (which corresponds to choosing the threshold $\theta$) and therefore $AUC$ acheives its maximum.

Equation 11 implies that:

$$
(\forall(p_n, p_a) \in \mathcal{N} \times \mathcal{A}) : S(p_n) < S(p_a)
$$

The reciprocal implication can be verified by taking $\theta = \max(\{S(p_n)|p_n \in \mathcal{N}\})$. This maximum exits since we suppose that $\mathcal{N}$ is discrete and finite. $\square$

The following is the proof of proposition 3 for the case $m = 2$.

*Proof.*
*The multi-prediction case:*

*Training:* First, we need to compute the predictions of the model when it achieves the global minimum. Indeed, the global minimum of the training loss is acheived for a model that predicts exactly the two normal possibilities: $\hat{p}_{t+1}^{(1)} = p_{t+1}^{(1)}, \hat{p}_{t+1}^{(2)} = p_{t+1}^{(2)}$, which minimizes the training loss since:

$$
\mathcal{L}_{NN} = \frac{1}{2} \sum_{p \in \{p_{t+1}^{(1)}, p_{t+1}^{(2)}\}} \min_{\hat{p} \in \{\hat{p}_{t+1}^{(1)}, \hat{p}_{t+1}^{(2)}\}} \|p - \hat{p}\| = 0
$$

The optimal set of predictions is therefore: $\{p_{t+1}^{(1)}, p_{t+1}^{(2)}\}$ (the uniqueness of the set can be easily demonstrated), Figure 1 in the main paper illustrate the optimal predictions.

*Anomaly scoring*: Once we computed the best model predictions, we can evaluate the anomaly detection performances. We denote by $S_{NN}$ the anomaly scores given by the multi-prediction model. We define the normal set $\mathcal{N} = \{p_{t+1}^{(1)}, p_{t+1}^{(2)}\}$ and the abnormal set $\mathcal{A} = \mathbb{R}^4 \backslash \{p_{t+1}^{(1)}, p_{t+1}^{(2)}\}$. We have:

$$(\forall p \in \mathcal{N}) : S_{NN}(p) = \min_{\hat{p} \in \{p_{t+1}^{(1)}, p_{t+1}^{(2)}\}} \|p - \hat{p}\| = 0$$

While:

$$(\forall p \in \mathcal{A}) : S_{NN}(p) = \min_{\hat{p} \in \{p_{t+1}^{(1)}, p_{t+1}^{(2)}\}} \|p - \hat{p}\| > 0$$

Therefore:

$$(\forall (p_n, p_a) \in \mathcal{N} \times \mathcal{A}) : S_{NN}(p_n) < S_{NN}(p_a)$$

We conclude by the lemma that $AUC = 100\%$.

*The single prediction case:*

*Training*: Given a single prediction model, the equation 3 in the main paper shows that the conditional expectation is the optimal solution. In the case of the synthetic dataset, it consists in the average of the next possible positions. Therefore the optimal prediction is $\hat{p}_{t+1}^{(s)} = \frac{p_{t+1}^{(1)} + p_{t+1}^{(2)}}{2} = (1, 0)$ (Figure 1 in the main paper). Note that this position is abnormal and that the optimal training loss is non-null:

$$\begin{aligned} \mathcal{L}_{single} &= \frac{1}{2} \sum_{p \in \{p_{t+1}^{(1)}, p_{t+1}^{(2)}\}} \|p - \hat{p}_{t+1}^{(s)}\| \\ &= \frac{(1-1)^2 + (1-0)^2 + (1-1)^2 + (-1-0)^2}{2} \\ &= 1 > 0 \end{aligned}$$

*Anomaly scoring:* We denote by $S_{single}$ the anomaly scores given by the single prediction model. We will show that:

$$(\exists (p_n, p_a) \in \mathcal{N} \times \mathcal{A}) : S_{single}(p_n) \geq S_{single}(p_a)$$

In fact, if we take $p_a = \hat{p}_{t+1}^{(s)} = (1, 0) \in \mathcal{A}$, we have:

$$\begin{aligned} S_{single}(p_a) &= S_{single}(\hat{p}_{t+1}^{(s)}) \\ &= \|\hat{p}_{t+1}^{(s)} - \hat{p}_{t+1}^{(s)}\| \\ &= 0 \end{aligned}$$

While if we consider: $p_n = p_{t+1}^{(1)} \in \mathcal{N}$:

$$S_{single}(p_n) = S_{single}(p_{t+1}^{(1)})$$
$$= \|p_{t+1}^{(1)} - \hat{p}_{t+1}^{(s)}\|$$
$$= (1-1)^2 + (1-0)^2$$
$$= 1 > 0$$

We conclude using the lemma (by contrapositive) that: $AUC < 100\%$.

*Finally, we have shown $AUC = 100\%$ is achieved only by the multi-prediction model trained via $\mathcal{L}_{NN}$.*

$\square$

**Proposition 4.** *Let $\bar{\mathcal{L}}_{naive}(f(X_M))$ the expected loss corresponding to $\mathcal{L}_{naive}(f(X_M), X)$ :*

$$\bar{\mathcal{L}}_{naive}(f(X_M)) = \mathbb{E}_{X \sim \mathbb{Q}} \left( \frac{1}{n} \sum_{k \in [\![1,n]\!]} \|f^{(k)}(X_M) - X\| \right) \qquad (12)$$

*In case of $L_2$ norm, the minimum expected loss is achieved by a MMP model: $f^* = (f^{*(k)})_{k \in [\![1,n]\!]}$ such that $(\forall k \in [\![1,n]\!]) : f^{*(k)}(X_M) = \mathbb{E}(X|X_M)$, which is similar to perform single prediction.*

*Proof.* We have:

$$\min_{f=(f^{(k)})_{k \in [\![1,n]\!]} \in \mathcal{F}^n} \bar{\mathcal{L}}_{naive}(f(X_M))$$

$$= \min_{f=(f^{(k)})_{k \in [\![1,n]\!]} \in \mathcal{F}^n} \mathbb{E}_X \left( \frac{1}{n} \sum_{k \in [\![1,n]\!]} \|f^{(k)}(X_M) - X\| \right)$$

$$= \min_{f=(f^{(k)})_{k \in [\![1,n]\!]} \in \mathcal{F}^n} \frac{1}{n} \sum_{k \in [\![1,n]\!]} \mathbb{E}_X(\|f^{(k)}(X_M) - X\|)$$

$$= \frac{1}{n} \sum_{k \in [\![1,n]\!]} \min_{f^{(k)} \in \mathcal{F}} \mathbb{E}_X(\|f^{(k)}(X_M) - X\|)$$

By definition of the conditional expectation $\mathbb{E}(X|X_M)$, we have $(\forall f^{(k)} \in \mathcal{F})$ for $L_2$ norm:

$$\mathbb{E}_{X \sim \mathbb{Q}}(\|\mathbb{E}(X|X_M) - X\|)$$
$$\leq \mathbb{E}_{X \sim \mathbb{Q}}(\|f^{(k)}(X_M) - X\|)$$

Therefore:

$$\frac{1}{n} \sum_{k \in [\![1,n]\!]} \min_{f^{(k)} \in \mathcal{F}} \mathbb{E}_X(\|f^{(k)}(X_M) - X\|)$$

$$= \frac{1}{n} \sum_{k \in [\![1,n]\!]} \mathbb{E}_{X \sim \mathbb{Q}}(\|\mathbb{E}(X|X_M) - X\|)$$

$$= \mathbb{E}_{X \sim \mathbb{Q}}(\|\mathbb{E}(X|X_M) - X\|)$$

Therefore, the minimum is achieved for $f^* = (f^{*(k)})_{k \in [\![1,n]\!]}$ verifying:

$$(\forall k \in [\![1,n]\!]) : f^{*(k)}(X_M) = \mathbb{E}(X|X_M)$$

$\square$

**Proposition 5.** *Let $\mathcal{X}$ the training dataset composed from normal samples. We denote by $\tilde{f} = (\tilde{f}^{(k)})_{k \in [\![1,n]\!]}$ a MMP model optimized via the nearest neighbor objective $\mathcal{L}_{NN}$ and achieving a loss $\mathcal{L}_{NN}(\tilde{f}(X_M), X)$ on a sample $X \in \mathcal{X}$. Let $\hat{f} = (\hat{f}^{(k)})_{k \in [\![1,n]\!]}$ a MMP model resulting from training the non-optimized predictors of $\tilde{f}$ via the non-participation loss $\mathcal{L}_{NP}$, and $\mathcal{L}_{NN}(\hat{f}(X_M), X)$ the loss of $\hat{f}$ on a sample $X$. We have $(\forall X \in \mathcal{X})$:*

$$\mathcal{L}_{NN}(\hat{f}(X_M), X) \leq \mathcal{L}_{NN}(\tilde{f}(X_M), X)$$

*Proof.* Since training the model with $\mathcal{L}_{NN}$ does not necessarily lead to optimizing all predictors (local minimum problem), the set of predictors that are not selected as nearest neighbors $\mathcal{U} \subset [\![1, n]\!]$ is not necessarily empty. Note that the demonstration also holds if $\mathcal{U} = \emptyset$.

By definition of $\mathcal{U}$, we have $(\forall X \in \mathcal{X})$:

$$\underset{k \in [\![1,n]\!]}{\arg\min} \|\tilde{f}^{(k)}(X_M) - X\| \notin \mathcal{U}$$

Therefore, we can re-write $\mathcal{L}_{NN}(\tilde{f}(X_M), X)$ as follows $(\forall X \in \mathcal{X})$ :

$$\mathcal{L}_{NN}(\tilde{f}(X_M), X) = \underset{k \in [\![1,n]\!] \setminus \mathcal{U}}{\min} \|\tilde{f}^{(k)}(X_{MP}) - X\|$$

Since the non-participation loss optimize only the predictors which belong to $\mathcal{U}$, the set of already optimized predictors $(\hat{f}^{(k)})_{k \in [\![1,n]\!]}$ obtained after training $(\tilde{f}^{(k)})_{k \in [\![1,n]\!]}$ using the non-participation loss $\mathcal{L}_{NP}$ remains the same:

$$(\forall k \in [\![1,n]\!] \setminus \mathcal{U}) : \hat{f}^{(k)} = \tilde{f}^{(k)}$$

Therefore, we have $(\forall X \in \mathcal{X})$ :

$$\begin{aligned}
\mathcal{L}_{NN}(\hat{f}(X_M), X) &= \underset{k \in [\![1,n]\!]}{\min} \|\hat{f}^{(k)}(X_M) - X\| \\
&\leq \underset{k \in [\![1,n]\!] \setminus \mathcal{U}}{\min} \|\hat{f}^{(k)}(X_M) - X\| \\
&= \underset{k \in [\![1,n]\!] \setminus \mathcal{U}}{\min} \|\tilde{f}^{(k)}(X_M) - X\| \\
&= \underset{k \in [\![1,n]\!]}{\min} \|\tilde{f}^{(k)}(X_M) - X\| \\
&= \mathcal{L}_{NN}(\tilde{f}(X_M), X)
\end{aligned}$$

Thus $(\forall X \in \mathcal{X})$ :

$$\mathcal{L}_{NN}(\hat{f}(X_M), X) \leq \mathcal{L}_{NN}(\tilde{f}(X_M), X)$$

$\square$

Table 7: MMP-AMS sub-networks number of parameters

| Sub-networks | Number of parameters |
|---|---|
| $f_F$ | 5.3M |
| $f_O$ | 5.3M |
| $f_C$ | 2.2M |
| $R$ | 460K |
| $P$ | 2.0M |
| Total | 15M |

## B  Implementation details

### B.1  Pseudo-code

In order to facilitate the reproducibility of our approach, we provide the pseudo-code 1 for training MMP-AMS.

### B.2  Network architectures

Regarding MMP-AMS, the architectures of $f_F$ and $f_O$ are made of two stacked auto-encoders, each one is composed of 4-strided convolutions with ReLU activations and strided transposed convolutions. The last layers are adapted for each modality and are followed by a sigmoid activation. We use a skip connection in the first auto-encoder to avoid the vanishing gradient issue. The hidden states $h_k$ are concatenated at the bottleneck of each auto-encoder to add the latent information. The architecture of $f_C$ is composed of 4 convolution layers followed by max-pooling with ReLU activation and 4 fully connected layers with a softmax activation. $f_C$ produces class vectors which have the same size as YOLOv3 class vectors. The first state predictor $P$, assembles a series of 4 convolution layers, a maxpooling and ReLU activations. The recurrent neural network $R$ is composed of 7 fully connected layers of hidden dimensions 256 with ReLU activations and a skip connection to avoid gradients vanishing. For all networks, we use a kernel size of 4x4 for convolutions and 2x2 for maxpooling. The architectures and the number of parameters of each module are shown in Figure 6 and in Table 7 respectively.

Concerning the location module, we use a small fully connected network which takes as input a class vector $X_C$ and predicts the means: $(\alpha_A(X_C))_{A \in \Theta}$ and standard deviations: $(\beta_A(X_C))_{A \in \Theta}$. The network is composed of 5 fully-connected layers with ReLU activation and a hidden size of 256. An illustration of the location module is shown in Figure 7.

### B.3  Implementation details on MNIST

In order to adapt our model to images, we mask an input image $I$ with a mask $M$. Then, we train a model $f_I$ to predict the multiple possible completions of the image $I$:

$$f_I(I \odot M) = (f_I^{(k)}(I \odot M))_{k \in [\![1,n]\!]}$$
$$= (\hat{I}^{(k)})_{k \in [\![1,n]\!]}$$

We use a similar architecture as for future frame prediction, and train the network using the same objective function:

$$\mathcal{L} = \mathcal{L}_{NN} + \lambda \mathcal{L}_{NP} \tag{13}$$

We train the network for 200 epochs for both datasets using Adam optimizer with a learning rate of $10^{-4}$ with a batch size of 128. We set $\lambda = 0.1$. We select predictors which have a participation below $\delta = 5\%$.

---

**Algorithm 1** MMP-AMS training pseudo-code

---

**Require:** number of predictors $n$, non-participation loss weight $\lambda$, selection threshold $\delta$, mask $M$, number of training epochs $n\_ep$, training dataset $\mathcal{X}$.
**Ensure:** Trained networks $(f_F, f_O, f_C, R, P)$
  Init $(f_F, f_O, f_C, R, P)$
  **for** $A \in \{F, C, O\}$ **do**
      Init the set of unoptimized predictors: $\mathcal{U}_A \leftarrow \emptyset$
      Init the participation counters: $c^{(A)} \leftarrow (0, .., 0) \in \mathbb{R}^n$
  **end for**
  **for** $i \in [\![1, n\_ep]\!]$ **do**
      **for** $X \in \mathcal{X}$ **do**
          # *Forward pass*:
          Apply mask: $X_{MP} \leftarrow X_P \odot M$
          Compute first state: $h_0 \leftarrow P(X_{MP})$
          **for** $k \in [\![1, n]\!]$ **do**
              Compute state: $h_k \leftarrow R(h_{k-1})$
              **for** $A \in \{F, C, O\}$ **do**
                  Compute prediction:
                  $\hat{X}_A^{(k)} \leftarrow f_A(X_{MP}, h_k)$
              **end for**
          **end for**
          # *Loss computation*:
          Init total loss: $\mathcal{L}_\Gamma \leftarrow 0$
          **for** $A \in \{F, C, O\}$ **do**
              Compute non participation loss:
              $\mathcal{L}_{NP} \leftarrow \sum_{k \in \mathcal{U}_A} \|\hat{X}_A^{(k)} - X_A\|$
              Compute nearest neighbor:
              $k_A^* \leftarrow \underset{k \in [\![1, n]\!]}{\mathrm{argmin}} \|\hat{X}_A^{(k)} - X_A\|$
              Compute nearest neighbor loss:
              $\mathcal{L}_{NN} \leftarrow \|\hat{X}_A^{(k_A^*)} - X_A\|$
              Update total loss:
              $\mathcal{L}_\Gamma + \leftarrow \mathcal{L}_{NN} + \lambda \mathcal{L}_{NP}$
              Update counter:
              $c_{k_A^*}^{(A)} + \leftarrow 1$
          **end for**
          # *Backward pass*:
          Backpropagate through $(f_F, f_O, f_C, R, P)$
      **end for**
      # *Update unoptimized predictors*:
      **for** $A \in \{F, C, O\}$ **do**
          Reset the set of unoptimized predictors:
          $\mathcal{U}_A \leftarrow \emptyset$
          Compute participations: $p^{(A)} \leftarrow c^{(A)}/\mathrm{sum}(c^{(A)})$
          **for** $k \in [\![1, n]\!]$ **do**
              **if** $p_k^{(A)} \leq \delta$ **then**
                  APPEND($\mathcal{U}_A, k$)
              **end if**
          **end for**
          Reset the participation counters:
          $c^{(A)} \leftarrow (0, .., 0) \in \mathbb{R}^n$
      **end for**
  **end for**

---

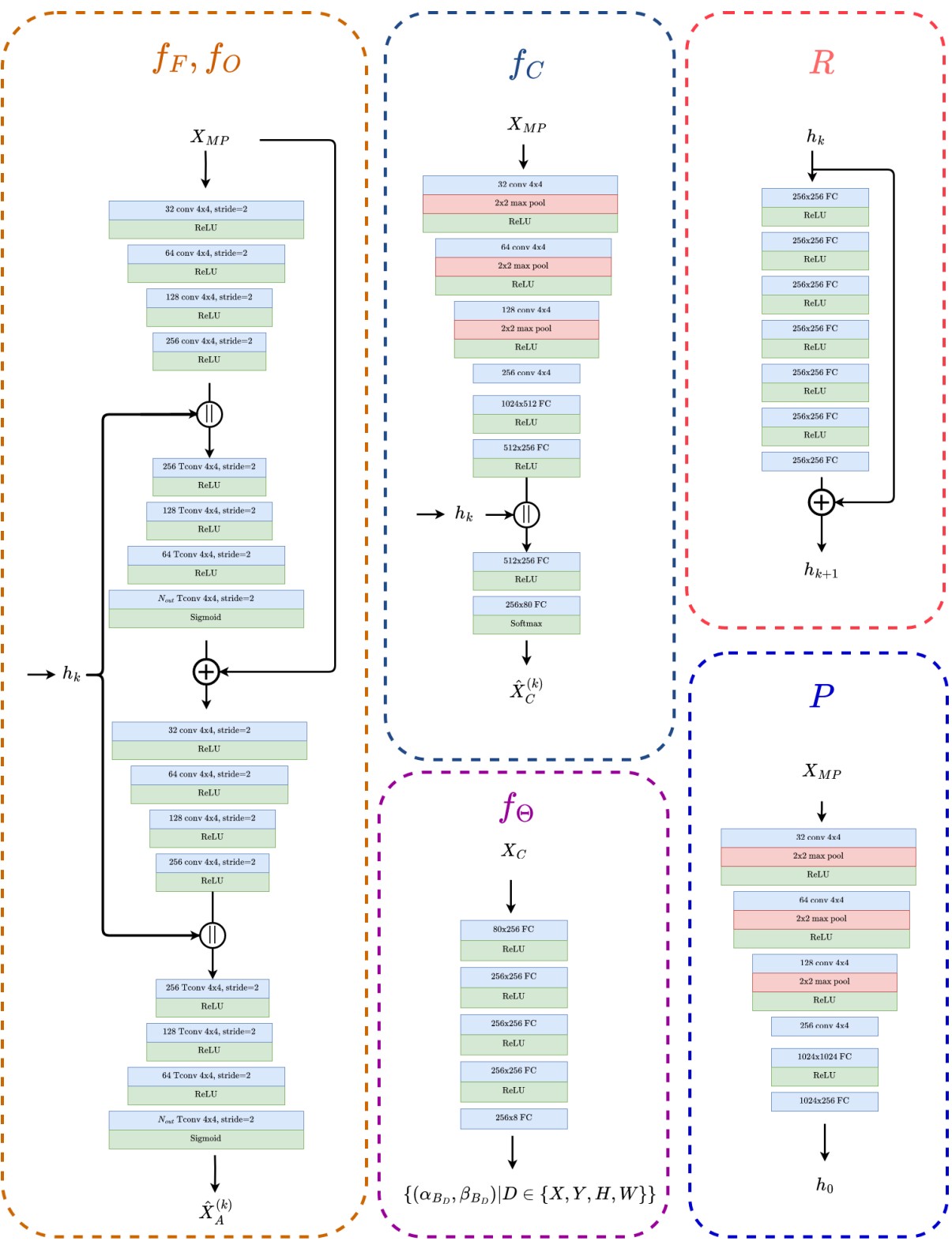

Figure 6: Overview of the architectures used for different method components: $f_F$: frame prediction network, $f_O$: optical flow prediction network, $f_C$: class prediction network, $R$: recurrent neural network, $P$: the first state predictor and $f_\Theta$: bounding boxes prediction network used for the location module. Notations: +: residual connection, ||: concatenation.

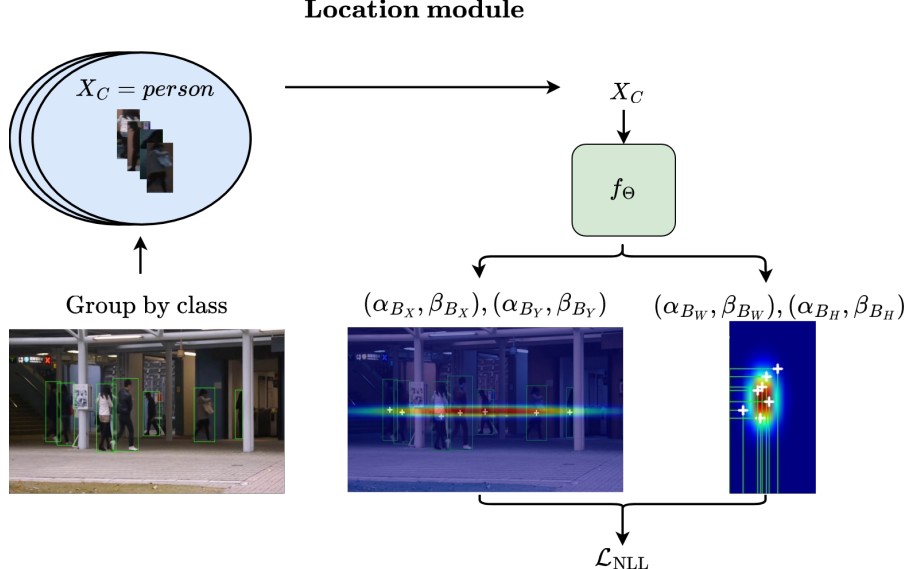

Figure 7: Overview of the the location module, we extract objects in the scene and group them by class. The class vector $X_C$ is given to $f_\Theta$ to predict the parameters of 4 Gaussians: $\mathcal{N}(\alpha_A(X_C), \beta_A(X_C))$ for $A \in \{B_X, B_Y, B_H, B_W\}$. We optimize the network via the negative log likelihood-loss $\mathcal{L}_{NLL}$.

Table 8: Comparison of the impact of non-participation loss on different baseline models.

(a) Impact on the multi-prediction model

| Model | $\mathcal{L}_{NP}$ | Mean prediction loss ($\downarrow$) |
|---|---|---|
| MP | $\times$ | 3.52 |
| MP | $\checkmark$ | **2.45** |

(b) Impact on the mixture of Gaussian model ($\times 10^{-3}$)

| Model | $\mathcal{L}_{NP}$ | Mean log-likelihood ($\uparrow$) |
|---|---|---|
| MoG | $\times$ | 2.41 |
| MoG | $\checkmark$ | **2.91** |

## C   Further experiments:

### C.1   Impact of the non-participation loss on baseline models:

In order to test the impact of the non-participation on different baselines, we evaluated it on two standard models that are used in the context of modeling multi-modal probability distributions: 1) a mixture of Gaussian model (MoG) trained via maximum log-likelihood 2) a multi-prediction (MP) model which consists of $n$ prototypes trained using the nearest neighbor loss ($\mathcal{L}_{NN}$). The dataset on which both models are trained consists of feature level representation of objects extracted from UCSDped2 using a pre-trained auto-encoder. Models are trained in a similar way, using Adam optimizer with a learning rate of $10^{-3}$ and a batch size of 64 for 150 epochs. For each of the two models, we assess the impact of adding the non-participation loss to training. For the Gaussian mixture model, the non-participation loss consists of maximizing the likelihood of samples w.r.t. Gaussians which do not maximize the likelihood for enough samples (the threshold $\delta = 5\%$ is used). As observed in the case of MMP models, using $\mathcal{L}_{NP}$ in training allows for a lower prediction loss in the case of the multi-prediction model (MP) as well as for higher likelihood for the mixture of Gaussians model (cf. Table 8).

### C.2   Impact of the temporal window on the performance of MMP-AMS on Avenue:

In order to test the impact of increasing the window size, we experimented MMP-AMS with larger prediction windows for future frames and the optical flows (cf. Table 9). As the predictions concern the spatio-temporal

window centred around the object, we consider the anomaly score for the full window. Nevertheless, since the same object can be detected multiple times in the same temporal window, we perform non-maximum suppression (NMS) to delete redundant detections. While increasing the time window results in better performance, it also requires additional computational costs and increases latency, as producing an anomaly score at time $t$ requires accessing the image a times $t + \text{window-size} - 1$.

Table 9: Influence of window size on anomaly detection performance on Avenue.

| Exp. | window size | RBDC | TBDC |
|------|-------------|------|------|
| e1 | 2 | 77.7 | 74.2 |
| e2 | 3 | 76.5 | 75.5 |
| e3 | 4 | 77.8 | 75.6 |
| e4 | 5 | **78.1** | **76.4** |

## D   Qualitative results

We show anomaly detection examples taken from the 3 datasets: ShanghaiTech, Avenue and UCSDped2: 10, 12, 11, 13, 14. The heatmaps illustrate object-level anomaly scores $S(X)$ and the graphs correspond to frame level scores $S(\mathcal{F}_t)$. We can see that our anomaly scores correlate well with the ground truth for different anomaly types including: unusual test objects (cf. Figures 13, 14,10) and unexpected behaviours (cf. Figures 11,12). Furthermore, we show in Figure 8 some frame predictions on test samples. Furthermore, we show an example of a case for which only the location aspect is abnormal, which is not annotated as an anomaly in UCSDped2 (Figure 9).

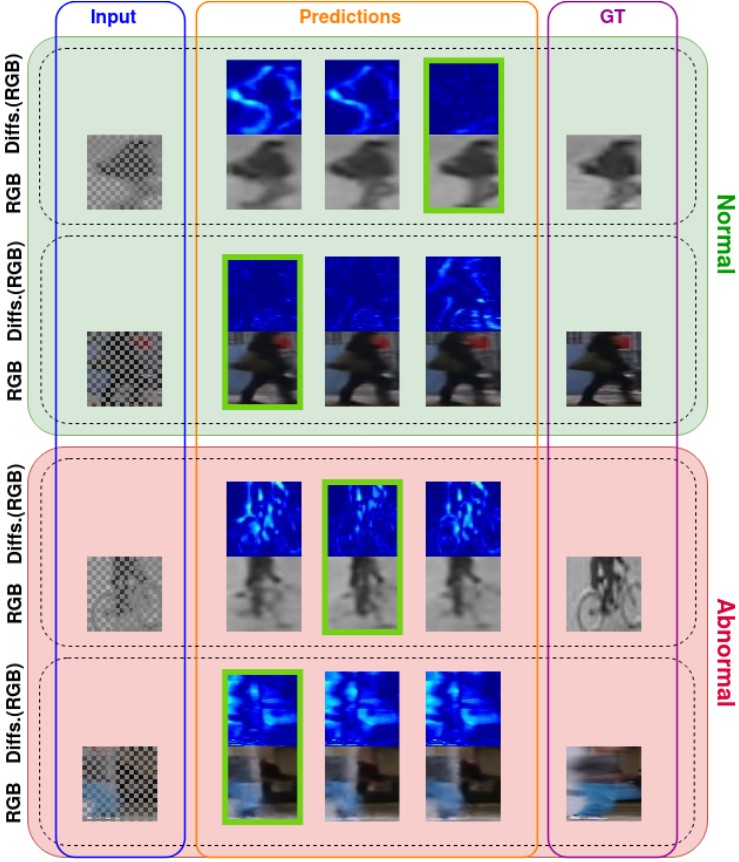

Figure 8: Visualization of normal and abnormal test examples. The first column indicates the masked input, the second column shows 3 predictions of the MMP-AMS framework. The nearest predictions are circled in green. The third column indicates the corresponding ground truth. We can observe that the nearest predictions are closer to the ground truth for normal samples than for abnormal samples.

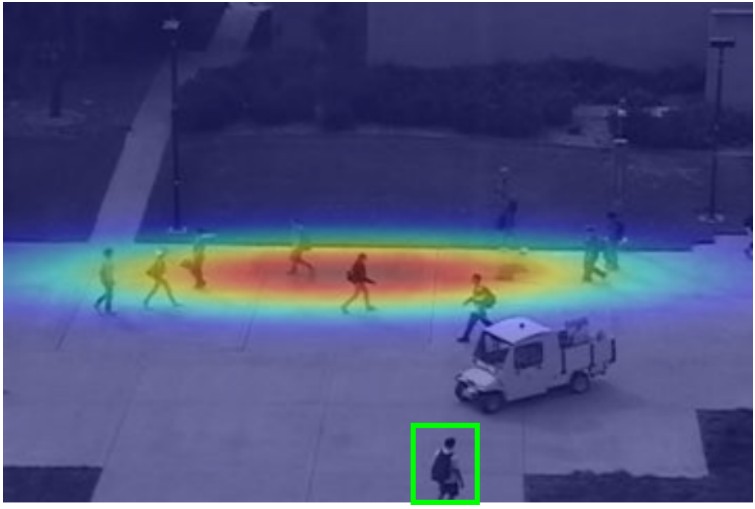

Figure 9: An example of a case for which only the location aspect is abnormal, which is not annotated as an anomaly in UCSDped2 (circled in green). The heatmap correspond to the density function learned by the location module.

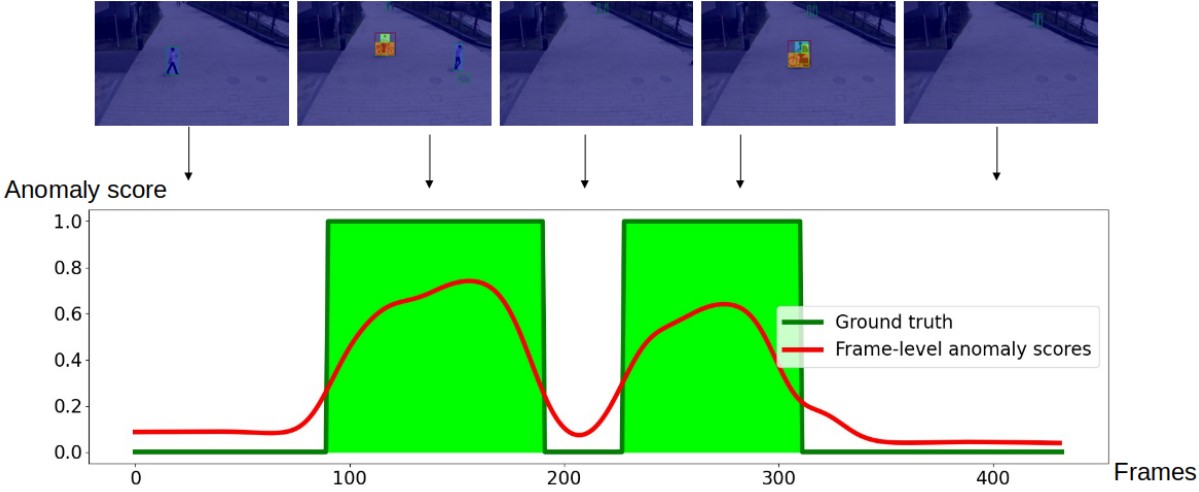

Figure 10: Qualitative results from video 01_0134 taken from ShanghaiTech dataset. The heatmaps illustrate object level anomaly scores $S(X)$ and the graphs correspond to frame level scores $S(\mathcal{F}_t)$. Red boxes indicate ground truth anomalies while green boxes indicate YOLOv3 detections. The anomaly consists of a person riding a bike.

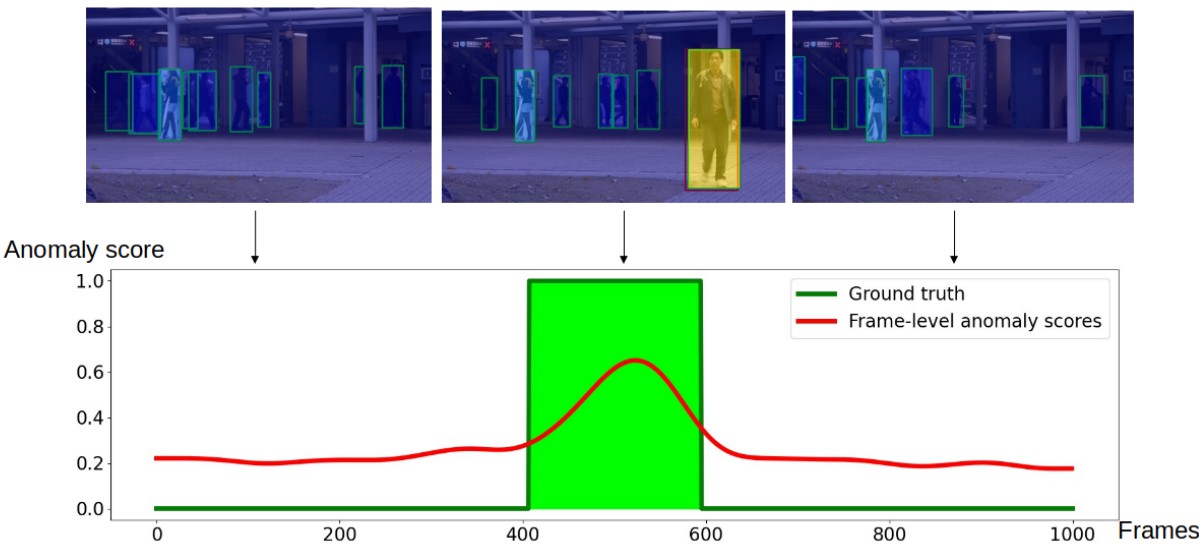

Figure 11: Qualitative results from video 15 taken from Avenue dataset. The heatmaps illustrate object level anomaly scores $S(X)$ and the graphs correspond to frame level scores $S(\mathcal{F}_t)$. Red boxes indicate ground truth anomalies while green boxes indicate YOLOv3 detections. The anomaly consists in a person walking towards the camera.

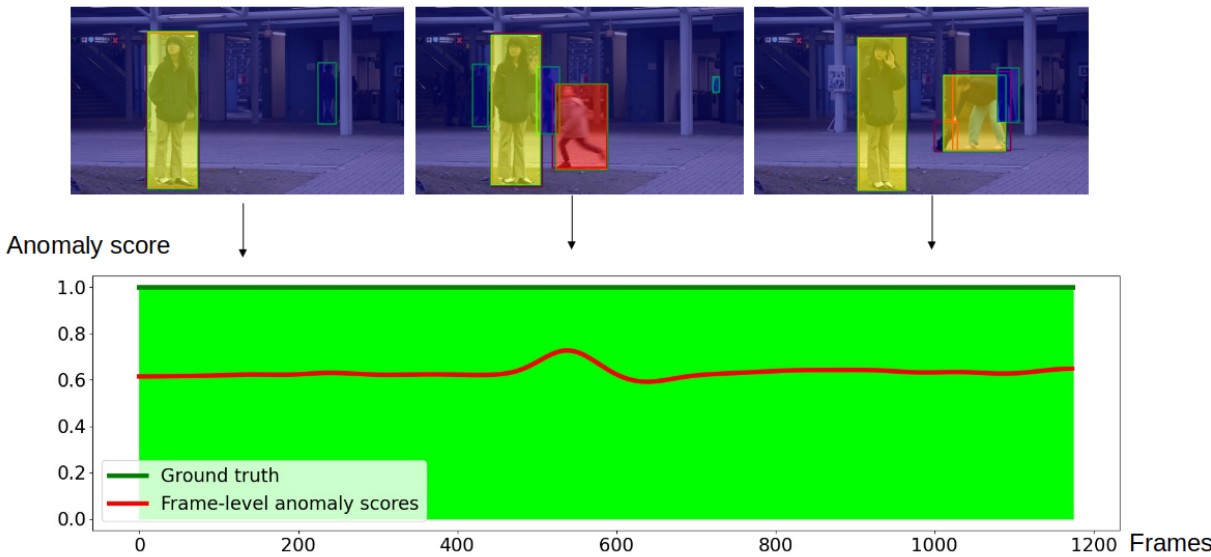

Figure 12: Qualitative results from video 09 taken from Avenue dataset. The heatmaps illustrate object level anomaly scores $S(X)$ and the graphs correspond to frame level scores $S(\mathcal{F}_t)$. Red boxes indicate ground truth anomalies while green boxes indicate YOLOv3 detections. The anomalies consist in a person standing in front of the camera, a kid jumping and a person throwing a bag

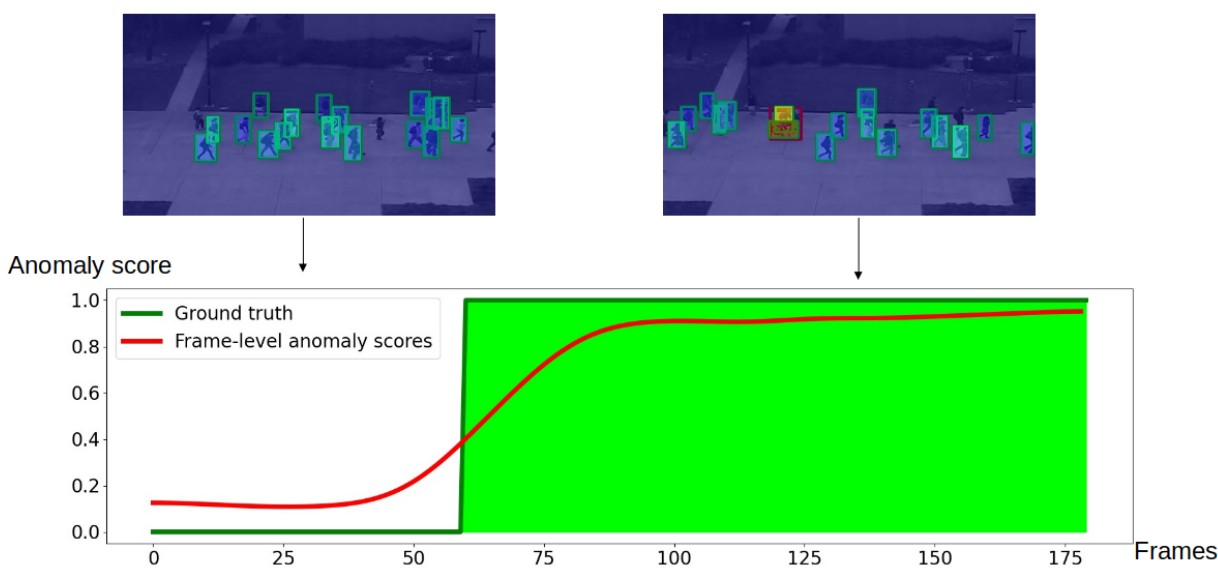

Figure 13: Qualitative results from video Test001 taken from UCSDped2 dataset. The heatmaps illustrate object level anomaly scores $S(X)$ and the graphs correspond to frame level scores $S(\mathcal{F}_t)$. Red boxes indicate ground truth anomalies while green boxes indicate YOLOv3 detections. The anomaly consists of a person riding a bike

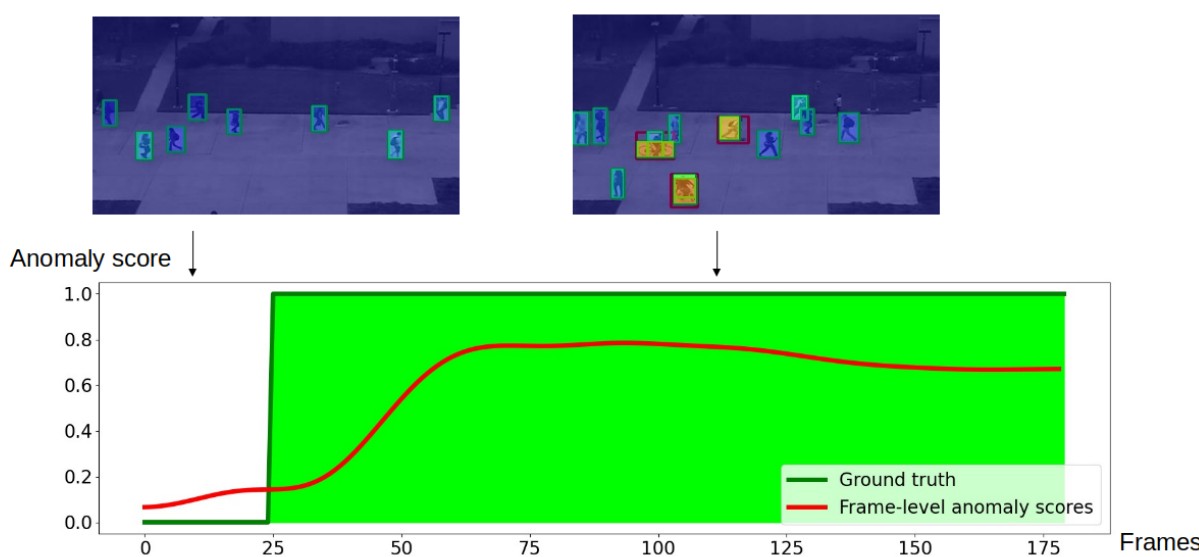

Figure 14: Qualitative results from video Test007 taken from UCSDped2 dataset. The heatmaps illustrate object level anomaly scores $S(X)$ and the graphs correspond to frame level scores $S(\mathcal{F}_t)$. Red boxes indicate ground truth anomalies while green boxes indicate YOLOv3 detections. The anomalies consist in skateboarding and riding a bike.

