# OpenReview forum: "Masked multi-prediction for multi-aspect anomaly detection"
_TMLR — Accepted by TMLR_

### Review · Reviewer_hQaC · 2024-04-02

**Summary Of Contributions:**

The paper introduces a novel approach called masked multiprediction (MMP) for anomaly detection in heterogeneous normal observations. MMP generates multiple likely normal outcomes, enhancing normality learning and improving anomaly detection performance. Additionally, the authors propose an adaptation of MMP, called MMP-AMS, which covers multiple aspects of normality such as appearance, motion, semantics, and location, leading to interpretable and modular anomaly detection. Experimental results validate the effectiveness of the proposed approaches.

**Audience:**

Yes

**Claims And Evidence:**

Yes

**Requested Changes:**

1. According to the suggestions, revise Section 1.
2. The experiments in  Section 3 should also evaluate on video datasets.

**Strengths And Weaknesses:**

Pros:
1. This paper is well-prepared, with comprehensive theoretical and experimental content.
2. The experimenst conducted on several benchmarks show the effectiveness of the proposed approach.


Cons:
1. The first paragraph of the Introduction section is too lengthy, while the specific background introduction is too brief, causing confusion for readers. Additionally, in this section, the motivation for the paper is not clearly articulated.
2. In the second paragraph of the Introduction section, the abbreviation "VAD" is used without its full expansion, which hasn't been introduced previously.
3. In Section 3, the experiments are all based on the MNIST dataset. However, since this paper focuses on video tasks, the disparity between real-world scenarios and MNIST is expected to be substantial.
4.What is the time complexity of the method proposed in this paper? Is it necessary to apply masking to each frame?
5. There is no open-source code available here.

---

> ### Author Response · Authors · 2024-05-05
> **Response to Reviewer hQaC**
>
> We are grateful to the reviewer for the constructive feedback and the valuable suggestions to improve our paper. We have taken all comments into account and added clarifications and experiences. Here are our responses to the weaknesses raised:
>
> > The first paragraph of the Introduction section is too lengthy, while the specific background introduction is too brief, causing confusion for readers. Additionally, in this section, the motivation for the paper is not clearly articulated.
>
> For the sake of clarity, we revised the introduction. We divided the first paragraph into three parts. In the first part, we introduce the problem of video anomaly detection. In the second part, we discuss existing approaches and detail previous work that is related to ours. In the third part, we revised the description of the motivation behind our proposed methodology. In addition, we revised Figure 1 to better illustrate our approach. We hope the new version will improve the readability.
>
> > In the second paragraph of the Introduction section, the abbreviation "VAD" is used without its full expansion, which hasn't been introduced previously.
>
> We introduced the term VAD at the beginning of the first paragraph for better clarity.
>
> > In Section 3, the experiments are all based on the MNIST dataset. However, since this paper focuses on video tasks, the disparity between real-world scenarios and MNIST is expected to be substantial.
>
> We carried out experiments on MNIST to illustrate the impact of masking. We have taken the suggestion into account by adding similar experiments (cf. Table 6 (revision)) to those conducted on MNIST. The empirical verification of propositions 2, 4 and 5 (cf. section 3) in the case of videos is provided in Table 4 (revision).
>
> > What is the time complexity of the method proposed in this paper?
>
> We added to the revised manuscript (cf. section 5.2) the time complexity for the whole pipeline, in addition to the inference time related to MMP-AMS alone. Using our implementation, the inference times of Yolov3 and FlowNet2 are 50 ms and 55 ms respectively. Since optical flow and object detection can run independently, they can be parallelized, in this case the whole pipeline can run in 13 FPS.
>
> > Is it necessary to apply masking to each frame?
>
> We verified experimentally the importance of masking on VAD benchmarks (a3 vs a4 in Table 5 and  e4 vs e7 Table 6.b (revision)) as well as on MNIST (e4 vs e7 in Table 2b). We apply masking in order to avoid that the model learns the trivial identity function (cf. section 3). Therefore, masking is applied to each frame.
>
> > There is no open-source code available here.
>
> Unfortunately, we cannot provide the code with an open-source licence as part of our activity is to transfer technologies to industrial partners by licencing the code we develop. Nevertheless, we have taken care to provide all the necessary details about the algorithm and its implementation for its reproducibility (cf. Section 5.2 in the main paper and Section B of supplementary material). We remain available to any researcher requiring further technical details.

---

### Review · Reviewer_feNb · 2024-04-03

**Summary Of Contributions:**

This paper proposes anomaly detection using a masked multi-prediction framework for modeling appearance, motion, and semantics (MMP-AMS).
In the MMP approach, the importance of multi-prediction is discussed. MMP recovers multiple original images using recurrent neural networks, and the minimum distance of multiple predictions is utilized. This loss is lower than in the single prediction case. Also, a non-participation loss that penalizes only the predictors that do not participate in training is proposed.
In the AMS approach, appearance, motion, and semantic aspects are used for the MMP approach in video anomaly detection. Experiments conducted on the UCSDped2, ShanghaiTech, and Avenue datasets demonstrate the effectiveness of MMP-AMS.

**Audience:**

Yes

**Broader Impact Concerns:**

No.

**Claims And Evidence:**

Yes

**Requested Changes:**

- Add a discussion of existing masked prediction methods.

- Enhance the description of the road dataset.

- Update the state-of-the-art results of VAD.

- Evaluate the loss of the MNIST dataset to confirm the Proportion 2, 5 in Sec. 3.

- Add the results of O, C to Table 4 to analyze the individual performance of these predictions.

- Add the conference or journal names for all references.

**Strengths And Weaknesses:**

Strengths
- The example of multi-prediction of the road scenario is intriguing.
- The authors theoretically discussed the multi- and single-prediction and the impact of loss functions.
- MMP-AMS is applied complex video anomaly detection.
- Most of the claims are supported and seem to be accurate.

Neutral
- Compared with existing studies on multi-prediction-based anomaly detection, this paper proposed masked multi-prediction.
This contribution itself seems to be not very innovative because there are existing studies on masked prediction for anomaly detection.

V.Zavtnik, M.Kristan, D.Skocaj, Reconstruction by Inpainting for visual anomaly detection, Pattern Recognition, vol. 112, 2011.

Weaknesses
- The discussion of the synthetic road dataset is not clear. P4.``We generated a synthetic dataset ‘'.
What is the training and test dataset? What is the anomaly? What is the model?
Obviously if the model predicts all anomal and normal sample correctly, even a single prediction can achieve 100% AUC. Thus, these conditions should be clarified.

- In the state-of-the-art results of VAD, the latest method is the year 2022, which is considered old.

- In the resutls of Table2, the location module significantly reduces performance on UCSDped2, and ShanghaiTech datasets.

---

> ### Author Response · Authors · 2024-05-05
> **Response to Reviewer feNb (Part 1)**
>
> We are grateful to the reviewer for the constructive feedback and the valuable suggestions to improve our paper. We have taken all comments into account and added clarifications and experiences. Here are our responses to the weaknesses raised:
>
> > compared with existing studies on multi-prediction-based anomaly detection, this paper proposed masked multi-prediction. This contribution itself seems to be not very innovative because there are existing studies on masked prediction for anomaly detection.
>
> We thank the reviewer for pointing out this reference, we add it to the related work. We would like to point out that we mentioned some prior works that used spatial masking for anomaly detection [1,2]. To better address the previous work in this field, we added a full paragraph in the related work concerning previous studies on this area. Thus, we updated the related work of the revised manuscript accordingly.
>
> Regarding the novelty of our contribution, our core motivation behind performing masked multi-prediction is to bridge the gap between prediction-based methods that predict poorly abnormal data but have difficulties to capture diverse normal patterns; and reconstruction-based methods that reconstruct well normal patterns but tend to over-generalise to anomalies. By performing masked multi-prediction, we leverage advantages from both families by learning diverse normal patterns thanks to multi-prediction and having a poor prediction of anomalies due to the masking constraint.
>
> Moreover, contrarily to existing masked anomaly detection studies which use either spatial or temporal masking, we proposed in MMP-AMS a joint spatio-temporal masking in order to learn the correlations between appearance and motion.
>
> [1] Self-supervised masking for unsupervised anomaly detection and localization. Chaoqin Huang, Qinwei Xu, Yanfeng Wang, Yu Wang, Ya Zhang
>
> [2] Ssmtl++:Revisiting self-supervised multi-task learning for video anomaly detection. Antonio Barbalau, Radu Tudor Ionescu, Mariana-Iuliana Georgescu, Jacob Dueholm, Bharathkumar Ramachandra, Kamal Nasrollahi, Fahad Shahbaz Khan, Thomas B. Moeslund, Mubarak Shah
>
>
> > The discussion of the synthetic road dataset is not clear. P4.``We generated a synthetic dataset ‘'. What is the training and test dataset? What is the anomaly? What is the model?
>
> We revised the description of the synthetic dataset (cf. section 3 (revision)), including the definition of normal and abnormal sets, as well as the description of the used model and the training procedure. In addition, we have also improved Figure 1 (revision). We hope that the revised version is clearer.
>
>
> > Obviously if the model predicts all anomal and normal sample correctly, even a single prediction can achieve 100\% AUC. Thus, these conditions should be clarified.
>
> Regarding the comparison of anomaly detection performances in Proposition 3, we aimed to show that a multi-prediction model achieving the minimum loss on the training set, achieves better anomaly detection performance than a single prediction model also achieving the minimal training loss on the same training set and evaluated on the same test set.
>
> > In the state-of-the-art results of VAD, the latest method is the year 2022, which is considered old.
>
>
> Regarding the results, as our approach is object-centric, for a fair comparison, we compared it with the state-of-the-art object-centric VAD methods using a standard evaluation protocol for those approaches, namely, we use the same baseline object detector (Yolov3) and compare the anomaly localization performance on the more recent anomaly localization evaluation metrics (RBDC and TBDC). To our knowledge, we included all the state-of-the-art methods that satisfy the same evaluation protocol.

---

> > ### Author Response · Authors · 2024-05-05
> > **Response to Reviewer feNb (Part 2)**
> >
> > > In the resutls of Table2, the location module significantly reduces performance on UCSDped2, and ShanghaiTech datasets.
> >
> > We explain the performance drop on UCSDped2 and ShanghaiTech by the fact that when we add the location module the model becomes able to detect location-related anomalies. However, those type of anomalies are not included in the definition of what is abnormal in UCSDped2 and ShanghaiTech. This means that samples, for which *only* the location aspect is abnormal (cf. Figure 3 in the supplementary material (revision)), are marked as normal by the ground truth for those datasets. Yet, since the location module detects those samples as abnormal, they are considered as an anomaly by the whole model (MMP-AMS w/ location module). Therefore, this lead to a false positive detection w.r.t the ground truth for those datasets, which explains the drop in performance. Thus, even if the location module detects location related anomalies well, we still can observe a drop in performance w.r.t the ground truth for those datasets. Therefore, as discussed in section 5, for a fair comparison with other methods that consider only the relevant aspects for those datasets (namely the appearance, motion and semantics), the location module should not be taken into account in those cases. This observation joins our discussion (section 5.4.2) on the importance of choosing the necessary and sufficient aspects of normality, in order to achieve satisfactory performance. Since our method models each aspect separately, it can be adapted to the aspects of interest.

---

### Review · Reviewer_cuob · 2024-04-14

**Summary Of Contributions:**

The paper presents a prediction-based anomaly detection method that integrates multiple outputs into masked representation learning to cope with heterogeneity. It employs a modified nearest neighbors loss and introduces different modules for multiple aspects of the anomaly. The work evaluates the method on three video benchmarks with comparisons to several previous approaches and ablation studies.

**Audience:**

Yes

**Broader Impact Concerns:**

Given its relevance to visual surveillance, it would be beneficial to include a concise discussion on its broader impact.

**Claims And Evidence:**

No

**Requested Changes:**

Please address the concerns in the weakness section.

**Strengths And Weaknesses:**

Strengths:

- The proposed multi-prediction multi-aspect approach is sensible for future prediction-based anomaly detection.
- The method achieves strong results on three benchmarks.

Weaknesses:

- The motivation is a bit unclear. First, the grouping of anomaly detection methods into explicit and implicit modeling lacks a clear justification. In particular, for video data, reconstruction and future prediction can be viewed as special cases of probabilistic methods under certain probability distribution assumptions (e.g. conditional Gaussian). Moreover, it is unclear why the paper focuses on the 'implicit' strategy. An 'explicit' model can naturally generate multiple outputs using multi-modal model distributions.

- The limitation of using a single prediction for anomaly prediction seems obvious and well-understood since the anomaly probability distributions can be multi-modal. The theoretical analysis in Sec 3.1 lacks sufficient insights.

- The design of the loss seems ad hoc by simply combining L_NN and L_NP with a fixed weight parameter $\lambda$. It is difficult to adjust the weight under different data instances. A more natural way is to match the predicted distribution of normality with the sampled groundtruth distribution, which has been explored in the uncertainty modeling literature.

- The overall technical novelty of the proposed strategy is incremental. The proposed loss is a minor extension of the NN loss from Guzman-rivera et al (2012); The masking strategy is widely used in representation learning; Using multi-aspect modeling is reasonable but it relies on three pre-defined aspects and thus has difficulty in generalizing to other unknown patterns.

- The experiments are a bit lacking in several aspects: 1) As the proposed method uses multiple subnetworks for capturing various aspects, it would be clearer to include the model sizes in the comparison for fairness. 2) For the comparison on the Avenue dataset, it would be more convincing for the proposed method to also use a longer temporal context. 3) The analysis/comparison seems to lack some simple baselines, such as a mixture of probabilistic models or a traditional multi-choice model on the learned feature space.

---

> ### Author Response · Authors · 2024-05-05
> **Response to Reviewer cuob (Part 1)**
>
> We are grateful to the reviewer for the constructive feedback and the valuable suggestions to improve our paper. We have taken all comments into account and added clarifications and experiences. Here are our responses to the weaknesses raised:
>
> > The motivation is a bit unclear. First, the grouping of anomaly detection methods into explicit and implicit modeling lacks a clear justification.In particular, for video data, reconstruction and future prediction can be viewed as special cases of probabilistic methods under certain probability distribution assumptions (e.g. conditional Gaussian). Moreover, it is unclear why the paper focuses on the 'implicit' strategy. An 'explicit' model can naturally generate multiple outputs using multi-modal model distributions.
>
> What we have considered “explicit” approaches are those that model the distribution of normal data or a transformation of it: $\mathbb{P}(f(X))$ where $f$ is the transformation applied to samples $X$, while we called "implicit" approaches those which learn some characteristics of normal data via a certain task such as future frame prediction or reconstruction for instance, those approaches generally learn a distribution in the form $\mathbb{P}(f(X)|g(X))$ where both $f$ or $g$ are transformations and $g(X)$ is informative about $f(X)$. We considered our approach as an "implicit" since it models a distribution in the form $\mathbb{P}(f(X)|g(X))$, where $f(X) = X$ and $g(X) = X_M = X \odot M$. In our case, $g(X)$ is informative about $f(X)$, as we don't mask totally $X$. However, if we consider the particular case when the masking is total $(M = [0]_{C\times H \times W})$, our method can also be seen as an "explicit" approach, in which case we model $\mathbb{P}(X)$ via a set of multiple predictions that are not conditioned on an input.
>
> Since $\mathbb{P}(f(X))$ is usually highly multi-modal, we focused on an "implicit" modeling from a practical perspective, since it allows for modeling less multi-modal distributions. Indeed, for the same transformation $f$, we know that adding an informative conditioning $g(X)$ on $f(X)$ reduces the uncertainty on $f(X)$, which formally can be written in terms of entropy as: $H(f(X)|g(X)) \leq H(f(X))$. Moreover, most SoTA object-centric VAD methods can be categorized as "implicit" [1,2,3,4,5]  which shows the effectiveness of the "implicit" modeling.
>
> In the revised version, we have replaced the terms "explicit" and "implicit" by defining each category as discussed above. We hope that the revised version is clearer (cf. Paragraph 2 in the section 1(revision)).
>
> [1] Ssmtl++:Revisiting self-supervised multi-task learning for video anomaly detection. Antonio Barbalau, Radu Tudor Ionescu, Mariana-Iuliana Georgescu, Jacob Dueholm, Bharathkumar Ramachandra, Kamal Nasrollahi, Fahad Shahbaz Khan, Thomas B. Moeslund, Mubarak Shah
>
> [2] Spatio-temporal predictive tasks for abnormal event detection in videos. Yassine Naji, Aleksandr Setkov, Angélique Loesch, Michèle Gouiffès, Romaric Audigier
>
> [3] Anomaly detection in video via self-supervised and multi-task learning. Mariana-Iuliana Georgescu, Antonio Barbalau, Radu Tudor Ionescu, Fahad Shahbaz Khan, Marius Popescu, Mubarak Shah
>
> [4] A Background-Agnostic Framework with Adversarial Training for Abnormal Event Detection in Video. Mariana-Iuliana Georgescu, Radu Tudor Ionescu, Fahad Shahbaz Khan, Marius Popescu, Mubarak Shah
>
> [5] A Hybrid Video Anomaly Detection Framework via Memory-Augmented Flow Reconstruction and Flow-Guided Frame Prediction. Zhian Liu, Yongwei Nie, Chengjiang Long, Qing Zhang, Guiqing Li
>
>
> > The limitation of using a single prediction for anomaly prediction seems obvious and well-understood since the anomaly probability distributions can be multi-modal. The theoretical analysis in Sec 3.1 lacks sufficient insights.
>
> We mentioned the limitation of using a single prediction, even though it seems intuitive, as it is commonly used in future prediction-based approaches for video anomaly detection (cf. paragraph 3 in related works (revision)). We revised section 3.1 by providing, in addition to the inequality, an expression for the prediction loss gap between multi-prediction and single prediction models, in terms of the distance between an optimal single prediction and optimal multiple predictions.

---

> > ### Author Response · Authors · 2024-05-05
> > **Response to Reviewer cuob (Part 2)**
> >
> > >The design of the loss seems ad hoc by simply combining $L_{NN}$ and $L_{NP}$ with a fixed weight parameter $\lambda$. It is difficult to adjust the weight under different data instances.
> >
> > We would like to point out that the non-participation loss has an important role since it allows for including all predictors in training and therefore achieving a lower prediction loss (Table 1,4 in the main paper (revision) and Table 2 in the supplementary material (revision)).
> >
> > We chose $\lambda$ parameter to have balanced values for both losses. Nevertheless, we conducted additional experiments to test the model sensitivity to $\lambda$ by trying several orders of magnitudes (cf. v5 to v8 in Table 1 (revision)). We observed slight variations in the prediction losses but with a full participation of all predictors for all tested values of $\lambda$. One possible explanation for the low sensitivity to $\lambda$ is that $L_{{NP}}$ optimizes parameters of predictors that have not been optimized by $L_{NN}$ (by definition). Consequently, if the predictors are independent the two losses optimize distinct sets of parameters. In addition, once all predictors start to be selected by $L_{NN}$, $L_{NP}$ is no longer used by definition.
> >
> > > A more natural way is to match the predicted distribution of normality with the sampled ground truth distribution, which has been explored in the uncertainty modeling literature.
> >
> > We thank the reviewer for the valuable suggestion. Indeed, our approach can be seen as a matching between a discrete distribution represented by the predictors  $(f^{(k)} (X_M))_{k \in [| 1, n |]}$ and the target distribution $\mathbb{P}(X|X_M)$. The nearest neighbor loss can be viewed as a "distance" between the two distributions. Indeed, we can formulate the training loss, given $X_M$, a MMP model $f$ and a set of samples from $\mathbb{P}(X|X_M)$: $\{X_i, i \in  [| 1, m |]\}$ as follows:
> >
> >
> > $
> > \sum_{i=1}^{m} \mathcal{L}_{NN}(f(X_M ),X_i )
> > $
> >
> > $
> > = \sum_{i=1}^{m} \min_{k \in [|1,n|]} || f^{(k)} (X_M ) - X_i  ||
> > $
> >
> > $$
> > = n \left( \min_{\gamma \in \mathcal{C}} \sum_{i=1}^{m} \sum_{k=1}^{n} \gamma_{i,k} || f^{(k)} (X_M ) - X_i  || \right)
> > $$
> >
> >
> > With the constraints $\mathcal{C}$ defined as:
> >
> >
> >
> >
> > $$
> > (\forall i \in [|1,m|]): \sum_{k=1}^{n} \gamma_{i,k} = \frac{1}{n}
> > $$
> > And,
> >
> > $$
> > (\forall i,k \in [|1,m|] \times [|1,n|] ): \gamma_{i,k} \in  \\{ 0,\frac{1}{n} \\}
> > $$
> >
> > This formulation is close to a discrete optimal transport problem  [1] with two main differences: there is no constraint on the marginal distribution over the predictions and the assignments are binary. The choice of modeling the target distribution via a discrete set of predictions has the advantage of being computationally and memory efficient.
> >
> >
> > [1] Discrete Optimal Transport: Complexity, Geometry and Applications. Quentin Mérigot & Édouard Oudet
> >
> > > The overall technical novelty of the proposed strategy is incremental. The proposed loss is a minor extension of the NN loss from Guzman-rivera et al (2012); The masking strategy is widely used in representation learning; Using multi-aspect modeling is reasonable but it relies on three pre-defined aspects and thus has difficulty in generalizing to other unknown patterns.
> >
> >
> > Regarding the novelty, our work is the first to the best of our knowledge, to apply multiple-choice learning in the context of VAD. Moreover, we have shown the importance of adding the proposed non-participation loss to the nearest neighbor loss in training (cf. Table 1,4 in the main paper (revision), Table 2 in supplementary material (revision)).
> >
> >
> > Regarding masking, unlike existing studies on masked anomaly detection that use either spatial or temporal masking, in MMP-AMS we have proposed joint spatio-temporal masking to learn correlations between appearance and motion.
> >
> >
> > The choice of predefined aspects in MMP-AMS was made in order to compare with previous approaches which use similar aspects to detect the corresponding anomalies. The choice of these aspects is application dependent and since our method models each aspect separately, it can be adapted to any aspects of interest.

---

> > > ### Author Response · Authors · 2024-05-05
> > > **Response to Reviewer cuob (Part 3)**
> > >
> > > > The experiments are a bit lacking in several aspects: 1) As the proposed method uses multiple subnetworks for capturing various aspects, it would be clearer to include the model sizes in the comparison for fairness. 2) For the comparison on the Avenue dataset, it would be more convincing for the proposed method to also use a longer temporal context. 3) The analysis/comparison seems to lack some simple baselines, such as a mixture of probabilistic models or a traditional multi-choice model on the learned feature space.
> > >
> > > 1) Regarding the model sizes, we were unable to find reported number of parameters of VAD approaches with which we compared our method. Nevertheless, we have added the model sizes for each subnetwork in MMP-AMS to allow comparison for future works (cf. Table 1 in the supplementary material (revision)). We have taken into account the size constraint of the model by choosing a recursive architecture with shared parameters which ensures that the model size remains constant as the number of predictions increases. Furthermore, we model the target distribution by a set of discrete predictions, which is both computationally and memory efficient.
> > >
> > > 2) We added experiments on Avenue with a longer temporal context. We observed a gradual increase in performance by increasing the temporal window for this dataset (cf. Table 3 in the supplementary material (revision)). While increasing the time window can result in a better performance, it also requires additional computational costs and increases the latency, as producing an anomaly score at time $t$ requires accessing the frame a time $t+ \text{window-size} - 1$.
> > >
> > > 3) We preformed additional experiments using two baseline models: 1) A mixture of gaussian model trained via maximum log-likelihood. 2) A multi-prediction model which consists on $n$ prototypes trained using the nearest neighbor loss. (cf. section C.1 in the supplementary material (revision)). We showed that the proposed $L_{NP}$ loss can improve training for those baselines also.

---

### Author Response · Authors · 2024-05-05
**General response**

We thank all reviewers for their constructive feedback and thoughtful comments. We have revised the manuscript on the basis of these feedbacks. The changes included in our revision are listed in the “Changes Since Last Submission” section. We respond to each reviewer individually below.

---

### Decision · Action_Editor_68rD · 2024-06-04

**Recommendation:** Accept as is

**Comment:**

This paper deals with video anomaly detection (VAD). The authors introduce a masked multi-prediction (MMP) approach based on combination of a nearest neighbour loss and a new non-participation loss. They also propose to extend the approach to model multiple normality aspects (MMP-AMS) and in order to refine anomaly detection. Theoretical results support the choices of the proposed framework, and preliminary experiments are provided on MNIST to justify the relevance of the proposed making approach over reconstruction methods on MNIST. Experiments are conducted on three real VAD datasets: UCSDped2, ShanghaiTech, Avenue datasets.

Initially, the reviewer appreciated the sensitivity of the approach, and acknowledge that most of the claims are validated by evidence. However, they also raised concern about the novelty and motivation behind the proposed approach, clarification of the positioning with respect to explicit vs implicit approaches, and justification of the different terms in the loss. They also question the experimental results, requesting clarifications on the toy dataset, requesting for more recent baselines, and a discussion on the impact of the location module. The authors' feedback was overall successful to answer to reviewer's concerns: the authors provided new experiments and modified the paper accordingly. After rebuttal, two reviewers recommended paper acceptance, while the third reviewer recommended rejection because they requested more recent baseline comparisons.

The AE carefully reviewed the submission and discussions. The AE considers that the approach is meaningful and that each contribution is validated by evidence, theoretically or experimentally. Although intuitive, the theoretical justification of multiple prediction will be of interest for the TMLR audience. The importance of MMP has been validated on MNIST but also on real video datasets, and the importance of leveraging diverse aspect for representing normality on the context of VAD is convincing. Therefore, the AE recommends paper acceptance.

**Audience:**

The paper addresses video anomaly detection (VAD), a very important problem in machine learning that will be of interest for a wide TMLR audience.

**Claims And Evidence:**

The claims are validated by evidence with a convincing set of theoretical and experimental justifications.